# Efficient neural spike sorting using data subdivision and unification

**Masood Ul Hassan**[1,2]*, **Rakesh Veerabhadrappa**[2], **Asim Bhatti**[2]*

**1** School of Engineering (Electrical and Renewable Energy), Deakin University, Waurn Ponds, Australia,
**2** Institute for Intelligent Systems Research and Innovation, Deakin University, Waurn Ponds, Australia

* m.ulhassan@deakin.edu.au (MUH); asim.bhatti@deakin.edu.au (AB)

## Abstract

Neural spike sorting is prerequisite to deciphering useful information from electrophysiological data recorded from the brain, in vitro and/or in vivo. Significant advancements in nanotechnology and nanofabrication has enabled neuroscientists and engineers to capture the electrophysiological activities of the brain at very high resolution, data rate and fidelity. However, the evolution in spike sorting algorithms to deal with the aforementioned technological advancement and capability to quantify higher density data sets is somewhat limited. Both supervised and unsupervised clustering algorithms do perform well when the data to quantify is small, however, their efficiency degrades with the increase in the data size in terms of processing time and quality of spike clusters being formed. This makes neural spike sorting an inefficient process to deal with large and dense electrophysiological data recorded from brain. The presented work aims to address this challenge by providing a novel data pre-processing framework, which can enhance the efficiency of the conventional spike sorting algorithms significantly. The proposed framework is validated by applying on ten widely used algorithms and six large feature sets. Feature sets are calculated by employing PCA and Haar wavelet features on three widely adopted large electrophysiological datasets for consistency during the clustering process. A MATLAB software of the proposed mechanism is also developed and provided to assist the researchers, active in this domain.

## Introduction

Neuro-engineering is an interdisciplinary research domain that provides a collaborative platform for engineers, scientists, neurologists and clinicians to grow a robust and reliable communication network between human brain and computers using advanced engineering procedures, methods, tools and algorithms [1–3]. It is largely accepted hypothesis that the brain passes information in terms of neurons' firings i.e. action potential or spikes over specific interval of time, known as neuron firing rate. Neurophysiological study of these hefty action potentials or spikes emanating from the neural network of the brain is essential to reveal the underlying behaviours and properties of neurons. A good understanding of the human brain neuronal network or nervous system is critically important in developing brain machine

**Data Availability Statement:** All relevant data are within the manuscript and its Supporting information files.

**Funding:** The research work is fully supported by Neural and Cognitive Systems Lab at Institute for

Intelligent Systems Research and Innovation, Deakin University". Although we do not have any explicit external funding grant linked to this work however the work is internally funded by the lab.

**Competing interests:** The authors have declared that no competing interests exist.

interfaces (BMIs), neuro-prosthetics and comprehensive brain-computer communication networks [4].

Electrophysiological analysis has attracted paramount importance, in recent years, in deciphering useful information about the underlying functional behaviour of the brain both in spontaneous and stimulated environments [5, 6]. This has paved the way of new discoveries in understanding the impact of external stimuli such as pharmaceuticals [7] and infections on the brain functionality [8]. Researchers have successfully developed the neural decoders from the neurophysiological study of intra neural recordings of human primary motor cortex to drive the artificial prostheses [9]. Electrophysiological studies also find significant importance in treating patients having neurological diseases or mental disorders especially in the case of epileptic disease. In addition, these studies have played vital role in understanding the gamma-protocadherine influences in regulating the neural network endurance and generating new neural synapses [10].

The significance of electrophysiological study of human brain lies in intercepting the neuronal signals with negligible interference in brain's natural functionality. Numerous electrophysiological methods are found in the literature to monitor the action potentials or spikes from neurons, such as intracellular glass pipette electrodes [11], patch clamp electrodes [12, 13], extracellular single or multi-site electrodes [14], and optical imaging devices [15, 16]. Among all, extracellular recordings using micro fabricated electrode arrays [17–19] are largely preferable in research because of its relatively less impact on the normal working behaviour of neurons [20]. Extracellular recordings are further categorised into invasive (in-vivo) and non-invasive (in-vitro) approaches [21]. In in-vivo approach, microelectrodes such as a probe or tetrode (probe with four electrodes) is surgically implanted in the understudy region of the brain. Whereas, in in-vitro approach, neurons are cultured on the separate dishes integrated with microelectrodes [22]. The neurophysiological technology implemented to record neural action potentials is very advanced, but still it is very immature to record the action potentials emanating from a single neuron. Brain consists of closely packed neurons that mostly excites simultaneously to encode information consisting of synchronised and correlated action potentials [23, 24]. Neurons present in the surrounding or neighbourhood of the understudy region, when excited, introduce noise in the neural recordings [25, 26]. Therefore, to study and analyse the behaviour of individual neurons and to group the action potentials having similar features into specific clusters, the concept of 'Spike Sorting' is implemented [27, 28].

An overview of in-vivo and in-vitro recordings and complete description of the steps involved in the spike sorting process is illustrated in Fig 1. Spike sorting consists of four main steps. First, raw data is filtered to minimise the effect of noise. The work of Choi et al. in [29] has significance importance in reducing the effect of background noise and detecting useful spikes trains from neural recordings at low signal to noise ratio (SNR) using multi resolution Teager energy operator (MTEO). Paralikar et al. in [30] proposed the virtual referencing (VR) method based on average functional electrode signal and inter-electrode correlation (IEC) method based on correlation coefficient between threshold exceeding spikes segments for common noise reduction. Common noise is generally produced by electromyographic activity, motion artifacts, and electric field pickup, especially in awake/behaving subjects. Pillow et al. in [30] proposed binary pursuit algorithm to significantly reduce the effect of stochastic background component of correlated Gaussian noise from the neural recordings. Takekawa et. al in [31] worked on filtering the biological noise from the neural recording using peak band pass filtering technique. Band pass filtering is a common practice among neural scientists for reducing the effect of background noise. This followed by spike extraction [32]. Abeles and Golstein in [33], elaborated extensively about multi-unit spikes detection. Threshold and inter-spike interval based detection methods are frequent and popular among researchers [34].

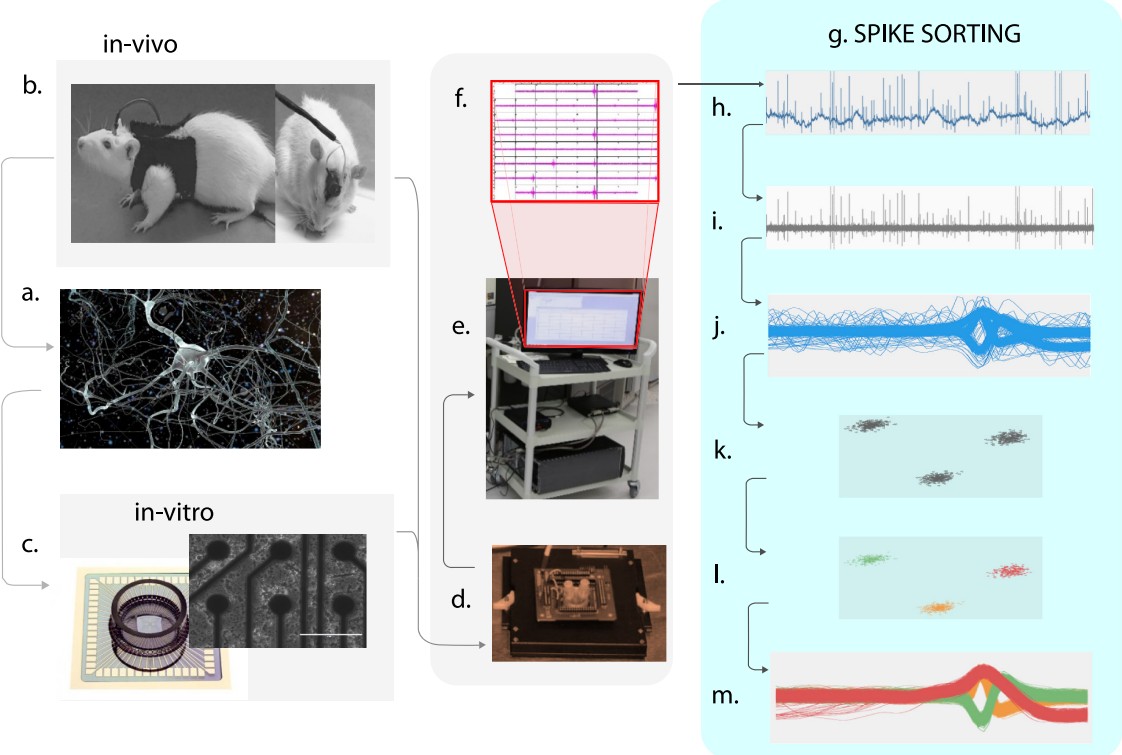

**Fig 1. An overview of spike sorting process with in-vivo and in-vitro recordings.** (a) Microscopic image of neural network in the brain. (b) Brains cells cultured on the Micro-Electrode Arrays (MEAs). (c) Implanted probe in the rat brain for in vivo recordings. (d) Data acquisition system to interface with MEAs (e) Computing machine for data processing and spike sorting. (f) Multichannel data acquisition and recording (g) Visualisation of complete spike sorting process. (h) Raw data after sampling and amplification. (i) Noise filtering of data using band pass filters. (j) Spikes detected using the threshold or inter-spike interval methods. (k) Feature extraction of the detected spikes to reduce the dimensionality of the data. (l) Clustered features after applying clustering algorithms extracted spike features. (m) Clustered spikes.

However, in the proposed algorithm the focus is on the computation efficiency of spike sorting algorithms rather than the spike estimation. For the proposed research work, spikes are extracted using labels provided with the data to make comparison of performance between different algorithms unbiased due to noise effects. The third step in spike sorting is the feature extraction of detected spikes [35]. The latest feature extraction technique is proposed by Zamani and Demosthenous in [36], however, feature extraction techniques that are largely practised by researchers are Principal Component Analysis (PCA) [37–39], Wavelet Transform [40–42] and Wavelet Packet Decomposition [43]. The last step in this process is the clustering of spikes into specific action potential groups having similar features [44]. For clustering, scientists have proposed numerous clustering algorithms in the literature [45–48] that are mainly classified into two main categories; Supervised [49] and Un-Supervised [50]. In supervised clustering, the number of clusters are predefined and the algorithms forced the spikes to fit into desired number of predefined clusters [51]. Whereas, in unsupervised clustering, algorithms, without having prior clustering information, automatically estimate the total cluster numbers and based on similarity in spike features, label the spikes into their respective groups [52]. The unsupervised clustering is more reliable and useful when there is no prior knowledge about clusters [53]. The spike sorting algorithms are mainly used offline and are implemented for behavioural quantification on pre-recorded neural datasets [54]. However,

researchers have developed online spike sorting algorithms that can quantify spike-clusters on live neural recordings [55]. The latest state of art in spike sorting process is presented in [56].

## Problem statement

Advancements in nanotechnology and nanofabrication has enabled neuroscientists and engineers to capture the electrophysiological activities of the brain at very high resolution, data rate and fidelity. However, to decipher useful information from these high dense electrode data, performance in terms of computational speed and accuracy of these spike-sorting algorithms, independent of their online and offline nature, plays an important role.

Stevenson and Kording in [57], presented data analysis issues due to progressive technological advancements of neural recordings. Progress in neural recording techniques enabling simultaneous multi channels recording is projected to double every 7 years resulting in high density and large size data. It is estimated that recording from 1000 neurons simultaneously could be achieved by 2025. The most recent automated spike sorting algorithm proposed by Chung et al. in [58] also highlighted the issue of low computational speed of spike sorting algorithms. Although they have proposed an efficient method for spike sorting, it lacks the speed researchers require for optimal results when sorting larger and high dense datasets. Wild et al. in [59] studied the performance evaluation on widely used clustering algorithms. His research outcomes highlighted the dependency of computational speed on data size or number of spikes to be clustered.

Chen and Cai in [60] investigated the issue and proposed that this behaviour is due to complexity of operations involved in the algorithms. They reported, for $n$ size of data, spectral clustering requires $O(n^2)$ (second order equation) operations in graph construction and $O(n^3)$ (third order equation) operations in Eigen-decomposition. These second order and third order equations prove the non-linear behaviour of spectral clustering. To motivate our analysis, spectral clustering was applied on five datasets of variable length and calculated the corresponding computational time as in Table 1. The plot in Fig 2, clearly depicts the non-linear behaviour in computational time required by spectral clustering to complete its operations with respect to data size.

The dependency of speed and computational time on data size in spike-sorting has made it very difficult to efficiently and accurately identify the total number of neurons in large and dense electrophysiological data. Furthermore, based on the work of Napoleon and Pavalakodi on large, dense and high dimensional breast cancer cell data [64], the accuracy of clustering algorithms is also somehow contingent to the data size. With the increase in data size the occurrence of false positives and negatives in spike sorting increases significantly, which reduces the overall efficiency and performance of the algorithms involved in the process.

Despite these challenges, in literature, researchers have developed numerous spike sorting algorithms to address the challenge of handling large and dense electrophysiological data. However, limited work has considered enhancing computational speed and efficiency by

**Table 1. Computational times of five datasets for spectral clustering.**

| Data Name | Data Size | # of classes | Computational Time |
|---|---|---|---|
| *MNIST* [61] | 70000 | 10 | 3654.90 |
| *LetterRec* [62] | 20000 | 26 | 195.63 |
| *PenDigits* [62] | 10992 | 10 | 60.48 |
| *Seismic* [63] | 98528 | 3 | 4328.35 |
| *Covtype* [62] | 581012 | 7 | 181006.17 |

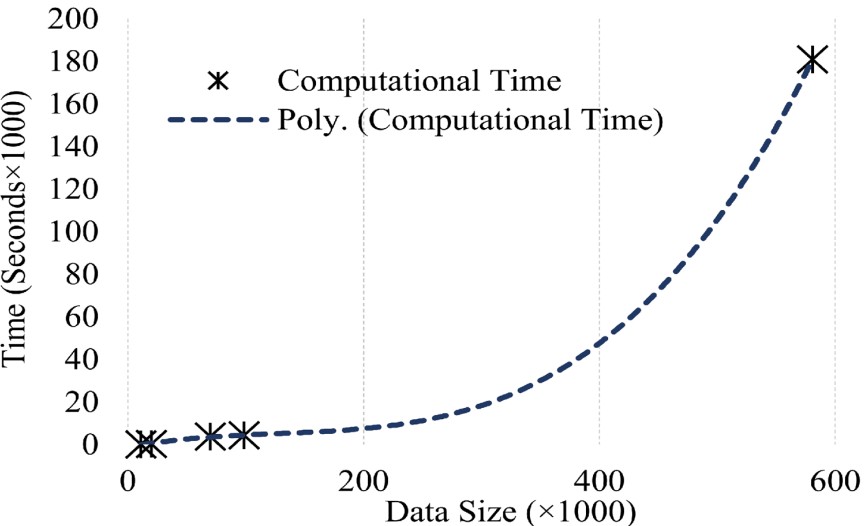

**Fig 2. Computational time versus data size plot.**

changing the way we input data into spike sorting algorithms. The proposed algorithm pre-processes data to significantly reduce computational time and to enhance speed and efficiency of a wide range of existing spike sorting algorithms. The proposed algorithm has great potential to be adopted by parallel computing approaches to further enhance spike sorting algorithms' efficiency for real-time online spike analysis.

## Proposed mechanism

The novelty of the proposed mechanism lies in its capability to operate the existing spike sorting algorithms at their peak efficiency by introducing the optimal length subsets of large electrophysiological data at clustering stage. The overall mechanism consists of three major steps as illustrated in Fig 3. 1) The first step involves subdivision of data into data-subsets of optimal length. The procedure to identify optimal length is discussed in next section. 2) The second step involves clustering spikes in data-subsets using conventional spike sorting algorithms. 3) The last step involves unification of the clustered subsets. The final unified clusters are then used to label the detected spikes representing complete large electrophysiological data into their respective neural classes. The comparison of conventional spike sorting and proposed algorithm is depicted in Fig 4. It is worth mentioning that the proposed mechanism deals with data- subdivision and unification to felicitate and enhance the performance of existing clustering algorithms and does not modify the internal workings of the algorithms employed in this study. A recently developed clustering algorithm "Mountainsort" by Chung et al. [58] uses a density based approach to cluster spikes can also be used with this mechanism for efficient spike sorting.

A similar approach of data subdivision is used by Pachitariu et al. in [65] for KiloSort algorithm. The algorithm divides the high dense neural data into small batches and uses them for mean-time processing of data filtering in the GPU that reduces the overall time of the spike sorting process. However, clustering of spikes is still deployed at complete large neural data-sets, which resulted into the slower computational speed of spike sorting at clustering stage. In addition, as opposed to proposed mechanism, the data-subdivision mechanism is limited to KiloSort and may not be applicable for other spike sorting algorithms. Furthermore, this algorithm failed to introduce the concept of optimal length for data-subdivision which is an

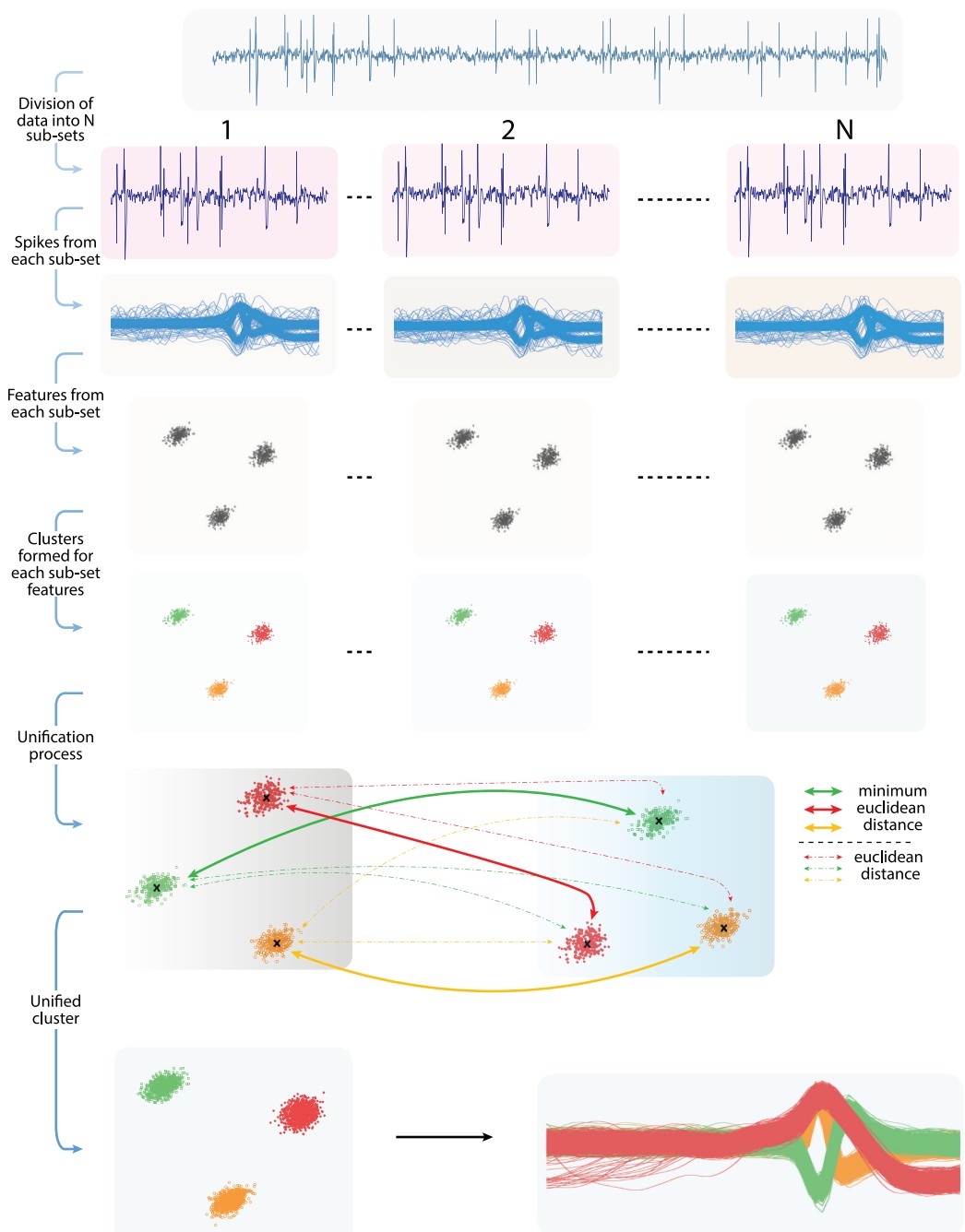

**Fig 3. Illustration of complete proposed mechanism.** The first step is to divide the large electrophysiological data into smaller groups. Second step involves the clustering of data-subsets using the conventional spike sorting algorithms. Last step involves the unification or merging of clustered data-subsets to get optimal clustering of complete large electrophysiological data.

important parameter to consider in enhancing the computational speed and operational cost of spike sorting process.

The detailed description of the steps involved in the proposed mechanism is provided in the following sections:

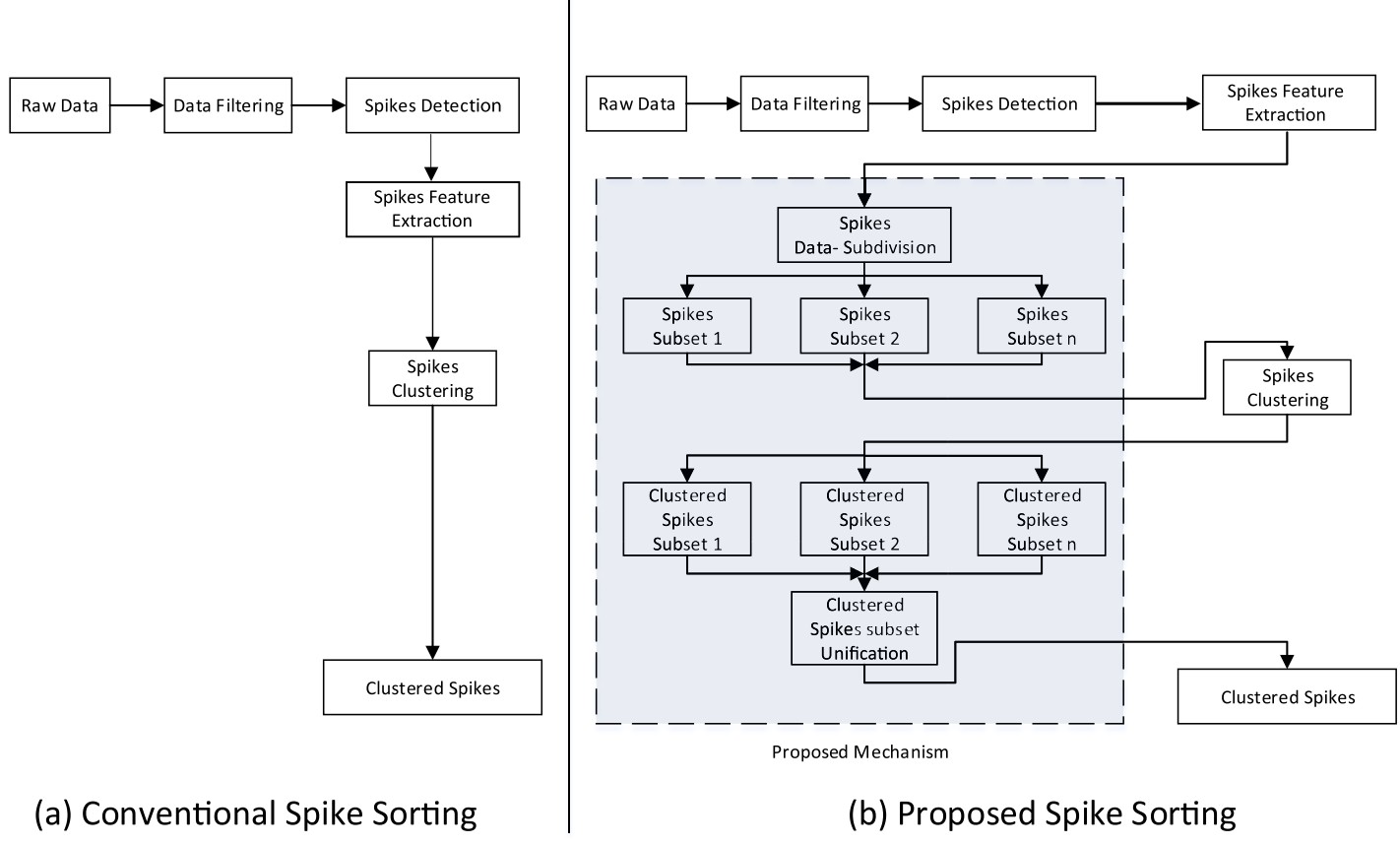

**Fig 4. Comparison of conventional and proposed spike sorting process.**

## Data subdivision

Subdividing large electrophysiological data into optimal length subsets is the most critical component of proposed mechanism. To form data subsets, let $D$ represents the electrophysiological data recorded at a single acquisition channel. The total number $N$ of optimal subdivisions is estimated as in Eq (1)

$$N = \frac{L}{O_L} \tag{1}$$

where $L$ is the length of data $D$ and $O_L$ is the optimal length for data-subsets. The procedure to calculate $O_L$ is presented in the next section.

The data-subsets are then estimated as in Eq (2).

$$S_d(n) = \begin{cases} D(1 + (n-1)*N) : D(n*N) & n*N < D_t \\ D(1 + (n-1) : D(D_t) & n*N \geq D_t \end{cases} \tag{2}$$

$$\forall\ n = \ 1,\ 2,\ 3,\ 4\ldots\ N$$

where $S_d(n)$ represents $n$ number of subdivided data-subsets of the large data $D$.

## Identification of optimal length ($O_L$) for data-subsets

$O_L$ is the range of values from which if the data size is selected to perform clustering, the clustering quality and computational efficiency of the conventional algorithms improve significantly. $O_L$ parameter is dependent on the algorithm type rather than on the data dynamics. Therefore it needs to be estimated only once for each algorithm. The $O_L$ parameter for ten commonly used clustering algorithms employed in this study, is estimated and shown in Fig 5b.

To understand the computational time vs data size behaviour, clustering is performed in an incremental manner. At every increment, the size or length of the data increases and the computational time is plotted with respect to data size as shown in Fig 5a. The size of data for which the clustering algorithm shows smoother behaviour is termed as $O_L$, that needs to be estimated for optimal clustering results.

In this research work $O_L$ is estimated by employing the work of Killick [66]. A threshold of 0.1 of the maximum rate of change of the computational time is used. The first change in computational time above the threshold is estimated to be the optimal length of the data-subset.

The procedure proposed in this research work to calculate $O_L$ is implemented on ten aforementioned commonly used clustering algorithms. The procedure is repeated hundred times to get an average $O_L$ value as an efficient measure for robustness in results. The calculated $O'_L s$ are depicted in Fig 5b. It is observed that the performance of clustering algorithms is independent of the data dynamics and feature extraction techniques. $O_L$ for all the algorithms adopted in this study, lies approximately in the same range for all three data and six feature sets, employed. Therefore, the computational performance of the algorithms depends on the length of the data set and not on the data dynamics.

Deviation of ($O_L$) from the estimated optimal point could lead to inefficient spike sorting performance. Data subdivision using optimal length is a compromise between computations involved in clustering process and unification process. ($O_L$) forms a direct relationship with clustering computations and an inverse relationship with computations involved in the unification process.

## Clustering of data

Data subdivision is followed by clustering of data-subsets employing conventional spike sorting algorithms. Ten algorithms, as illustrated in Fig 5, are employed in this study due to their wide adaptability in spike sorting research. The algorithm proposed is independent of the clustering procedure; therefore, any other clustering technique could be adopted in this mechanism.

## Unification of subclusters

After the clustering is performed on each data-subset, the unification of the sub clusters is performed. Sub-clusters are unified by identifying the overlap between the bounded regions of sub-clusters. The bounded region (BR) is a '$m$' dimensional set which consist of minimum and maximum variations of '$m$' dimensional spike feature waveforms in each dimension for a corresponding sub cluster. The bounded region for $j^{th}$ sub cluster is given by relationship in Eq (3).

$$BR_{j,i} = \left\{ \begin{bmatrix} min \\ max \end{bmatrix}_{j,1} \begin{bmatrix} min \\ max \end{bmatrix}_{j,2} \begin{bmatrix} min \\ max \end{bmatrix}_{j,3} \cdots \begin{bmatrix} min \\ max \end{bmatrix}_{j,m} \right\} \tag{3}$$

**a)** Computational time vs data size

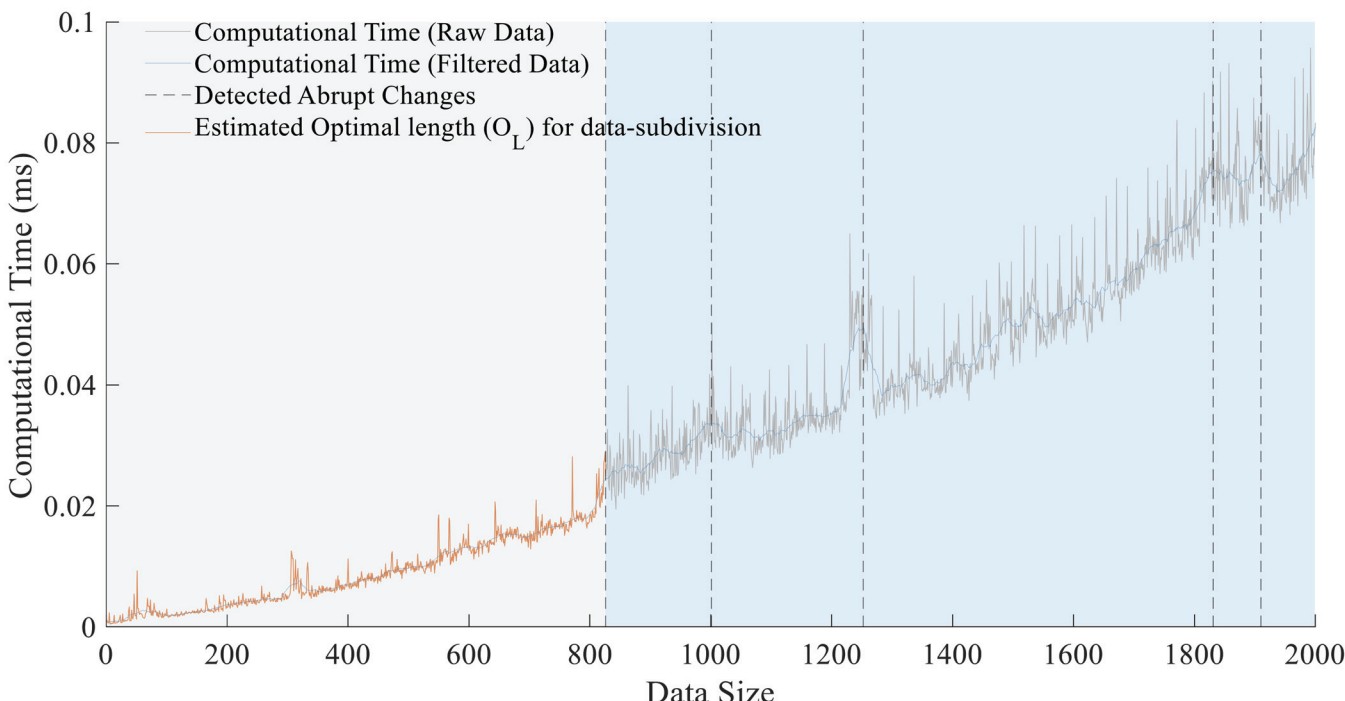

**b)** Algorithms optimal data length range for efficient spike sorting

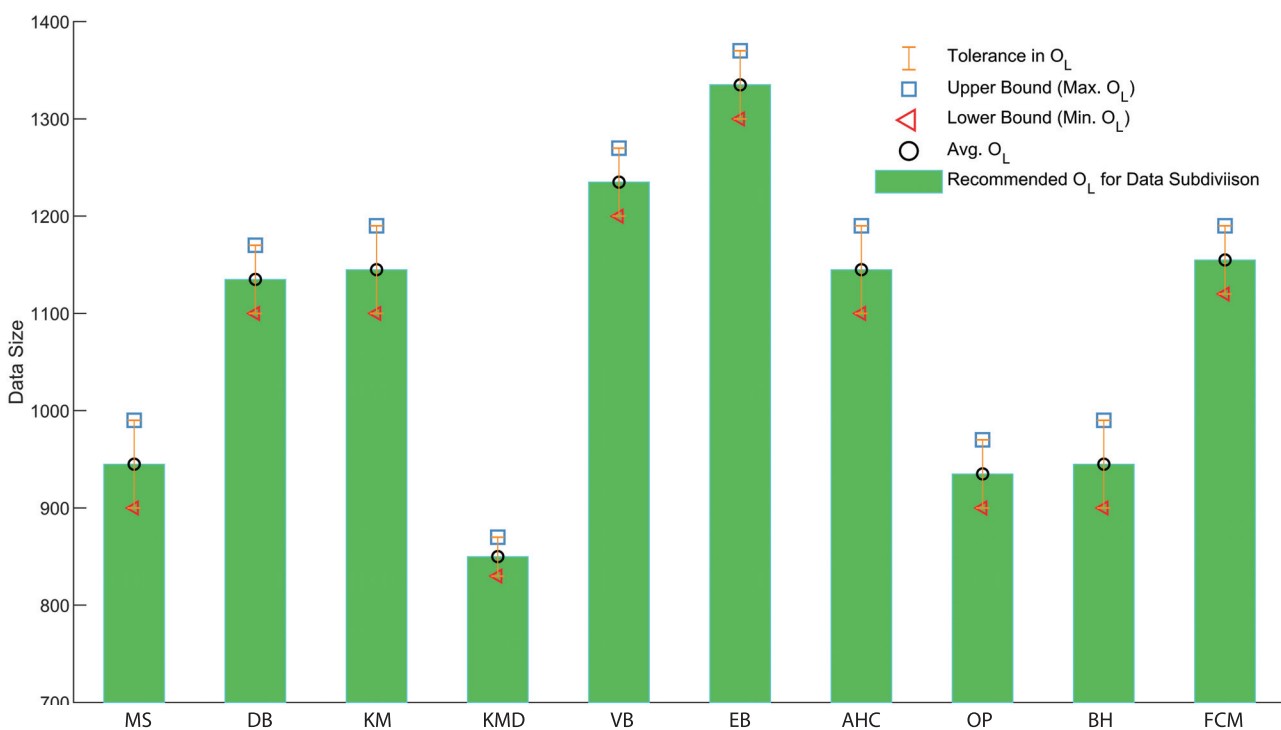

**Fig 5. Identification of optimal length $O_L$.** (a) Illustrates the description of steps involved in identifying the $O_L$ for spike sorting algorithms. Computational time versus data size plot. The X-axis shows the length of the data increasing from zero to 2000 while the Y-axis shows the corresponding time taken by the clustering algorithm to perform clustering process, in milliseconds. The computational time is the processing time after *movmean* filter(20 datapoints length) filtered the unwanted ripples in the plot and returned smooth curves. Detected abrupt changes in the plot taking 0.1 of the maximum rate of change in computational time as threshold (d) Identified optimal length $O_L$ of data subsets used for data subdivision. b) Optimal Length ($O_L$) for ten commonly used clustering algorithms. The average value over ten repetitive analyses is given as robustness of the measure in optimal length for data subdivision.

Where $BR_{j,i}$ is the '$m$' dimensional bounded region for $j^{th}$ sub cluster with $j \in [1, 2, 3,.., k]$ and '$k$' is the total number of sub clusters participated in the unification process. $\begin{bmatrix} min \\ max \end{bmatrix}_{j,i}$ are the minimum and maximum variations of spike feature waveforms for $j^{th}$ sub cluster and in $i^{th}$ dimension and $i \in [1, 2, 3, ..., m]$.

In this study, since 10 PCA or 10 Wavelet features are used to transform the spike waveform into spike feature waveform. So '$m$' is 10 in this particular case and BR is a 10 dimensional set with minimum and maximum values providing variation of spike feature waveforms in each dimension for a particular sub cluster.

The bounded region is calculated for all '$k$' sub clusters participated in the unification process. The sub clusters, having overlapping bounded regions in all dimensions, are unified together. The unification process for a 2 dimensional sub clusters is shown in Fig 6. In the Fig 6, it is also illustrated how sub clusters unify in three different scenarios i.e. 1) no overlapping region between sub clusters 2) overlap between two distinct sub clusters and 3) multiple overlapping sub clusters.

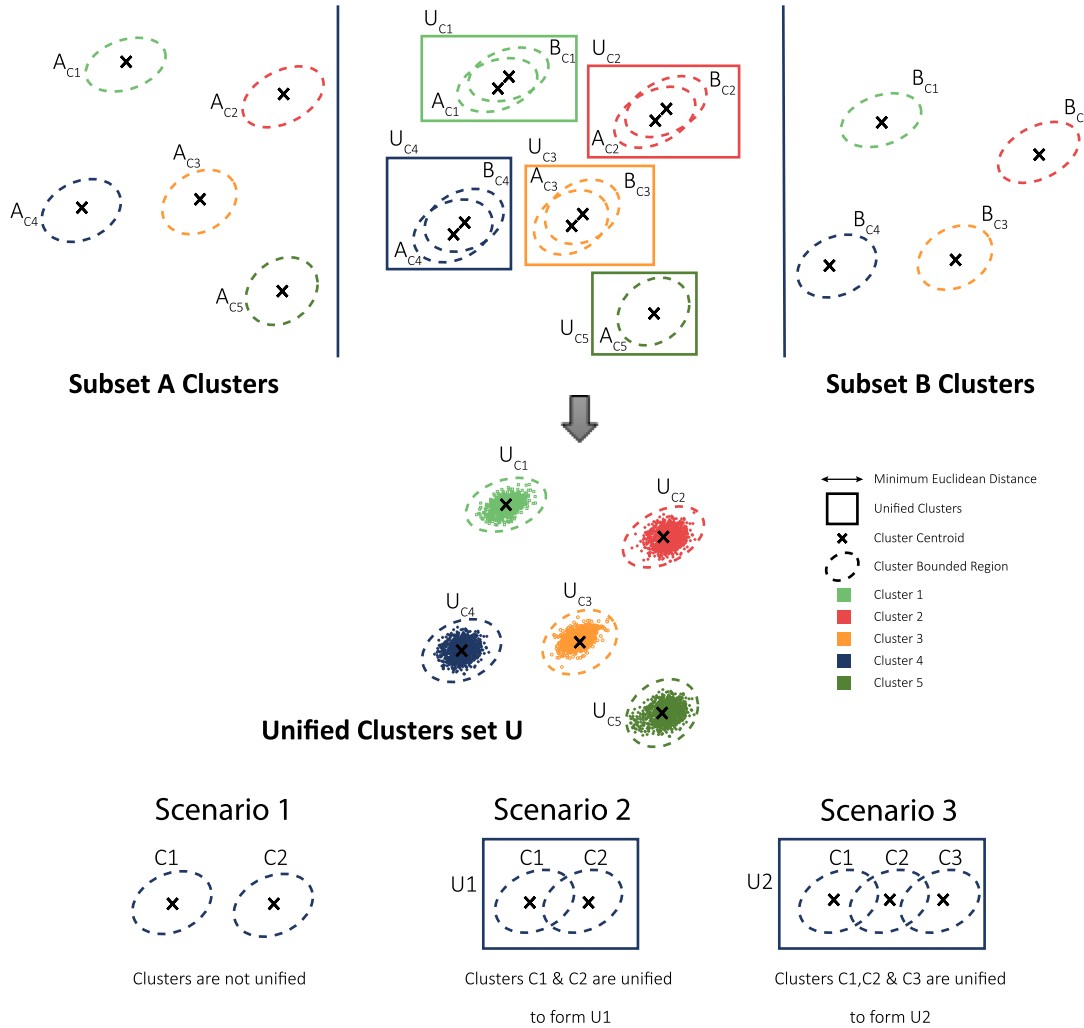

**Fig 6. Mechanisim to unify or merge clusters.**

To eliminate the impact of outliers in deciding the bounded region for unification process, the spike feature waveforms are filtered in each sub clusters. The filter proposed in this study is based on Euclidean distance. A sub cluster having '$m$' dimensional spike feature waveforms, should have an '$m$' dimensional centroid '$C$'. It is important to note that, the complete '$m$' dimensional spike feature waveform is considered as a single point in '$m$' dimensional space in calculating the Euclidean distance. Therefore, for each spike feature waveform, an Euclidean distance from spike feature waveform to its sub cluster centroid is calculated. The relationship to calculate the Euclidean distances is given in Eq (4).

$$ED_l(C_i, S_{l,i}) = \sqrt{(C_1 - S_{l,1})^2 + (C_2 - S_{l,2})^2 + (C_3 - S_{l,3})^2 + \cdots + (C_m - S_{l,m})^2} \qquad (4)$$

$ED_l(C_i, S_{l,i})$ is the Euclidean distance calculated for the $l^{th}$ spike feature waveform $S_{l,i}$ and sub cluster centroid $C_i$. $l \in [1, 2, 3, 4 \ldots, n]$ and $i \in [1, 2, 3, \ldots, m]$ where '$n$' is the total number of spikes in a sub cluster and '$m$' is the spike feature waveform dimension.

Since, the Euclidean distance is calculated based on Eq (4), a $n \times 1$ Euclidean distance matrix (EDM) is generated, as in Eq (5).

$$EDM = \begin{bmatrix} ED_1 \\ ED_2 \\ ED_3 \\ \vdots \\ ED_n \end{bmatrix} \qquad (5)$$

This *EDM* matrix is used to identify outliers in spike feature waveforms. From *EDM*, a Mean '$\mu$' and Standard Deviation '$\sigma$' is calculated by using Eqs (6) and (7).

$$\mu = \frac{\sum_{t=1}^{n} ED_t}{n} \qquad (6)$$

$$\sigma = \sqrt{\frac{1}{n} \sum_{t=1}^{n} (ED_t - \mu)^2} \qquad (7)$$

Using the mean and standard deviation of a normal distribution curve, the Euclidean distance values are converted into Z scores using the Eq (8).

$$Z_l = (ED_l - \mu)/\sigma \qquad (8)$$

The Z score distribution determined by Eq (8) is then used to identify the data outliers in the EDM matrix given by Eq (5). To this aim, we considered two scenarios; 1) when the Z score distribution of the *EDM* matrix is normal and 2) when the Z score distribution of the *EDM* matrix is skewed. There are numerous methods that can determine the normality of the data distribution as in [67–69]. However, in this study, the normality of the Z score distribution of *EDM* matrix is determined using the Interquartile Range IQR method [70].

The quartiles are three points that divide the data set into four equal groups, each group comprising a quarter of the data, for a set of data values which are arranged in either ascending or descending order. Q1, Q2, and Q3 are represent the first, second, and third quartile's value. The Interquartile Range (IQR) is basically a difference between the first quartile (Q1) and

third quartile (Q3). The IQR of the Euclidean distance matrix sorted in ascending order can be determined using relation given in Eq (9).

$$IQR = Q3 - Q1 \tag{9}$$

Where $Q1$ is first quartile and it is the median of lower half of the euclidean distances sorted in ascending order and $Q3$ is the third quartile and it is the median of upper half of the euclidean distances sorted in ascending order.

If the distance of the $Q1$ and $Q3$ from the median of the complete dataset containing Euclidean distances is equal, the data is normally distributed and the bell shaped curve is symmetric. If the distance from data mid-point to $Q1$ is bigger than $Q3$, the data distribution is skewed towards left, and if $Q3$ is bigger than $Q1$, the data distribution is skewed towards right.

For a normal distribution of the data, when bell shaped curve is symmetric, Empirical rule is valid and the outlier filter (OF) is is defined as a range between $\mu \pm 2\sigma$ and its is given by Eq (10).

$$OF = \begin{bmatrix} min & max \end{bmatrix} = \begin{bmatrix} \mu - 2\sigma & \mu + 2\sigma \end{bmatrix} = \begin{bmatrix} -2Z & 2Z \end{bmatrix} \tag{10}$$

For a nonsymmetric or left and right skewed distributions, 1.5 Interquartile Range (1.5 *IQR*) filter is used to identify the sub cluster outliers. The factor 1.5 is empirically derived and being used by novel researchers in statistics for skewed data to identify outliers [71, 72]. Therefore, in this study 1.5 *IQR* based outlier filter (OF) is designed to remove data outliers in skewed distribution and it is given by Eq (11).

$$OF = \begin{bmatrix} min & max \end{bmatrix} = \begin{bmatrix} Q1 - 1.5 \times IQR & Q3 + 1.5 \times IQR \end{bmatrix} \tag{11}$$

All the featured spikes, having the Euclidean distance lies within the OF range, are considered in estimating the bounded region in Eq (3) for unification of sub clusters.

A similar approach is adopted by Aksenova et al. in [73] to perform training of online spike sorting algorithm employing phase space. Their algorithm is focused on efficient noise reduction rather than optimisation of computational efficiency.

## Performance evaluation of the proposed algorithm

In this research work, the performance of the proposed algorithm is evaluated using two indicators, computational time and clustering quality. A comparative performance of the proposed algorithm with respect to the conventional algorithm is presented in Fig 7(a) and 7(b).

For validation, ten most widely adopted clustering algorithms are employed in the proposed research work. The algorithms include MeanShift (MS) [74], Density-based spatial clustering of applications with noise (DBSCAN) [75], Kmeans (KM) [76], Kmedoids (KMD) [77], Fuzzy C means (FCM) [78], Variational Bayesian Gaussian Mixture Model (VBGMM) [79], Expectation Maximization Gaussian Mixture Model (EMGMM) [80], Agglomerative Hierarchical Clustering (AHC) [81], Birch (BH) [82] and Ordering Points to Identify the Clustering Structure (OPTICS) [83].

To quantify computational efficiency of the proposed algorithm, three data sets are used, reported by Quiroga [84], because of their wider adoptability and ground truth availability. These datasets includes two (2) simulated Dataset 1 (D1) and Dataset 2 (D2) and one human Dataset 3 (D3). Human data is originated from multiunit recording in the temporal lobe of an epileptic patient from Itzhak Fried's lab at UCLA [84]. The information regarding spatio-temporally overlapping spikes as a result of multi-unit recordings can be identified using "Matching Persuit" algorithm [85]. However in this study, the multi-unit spikes are already detected

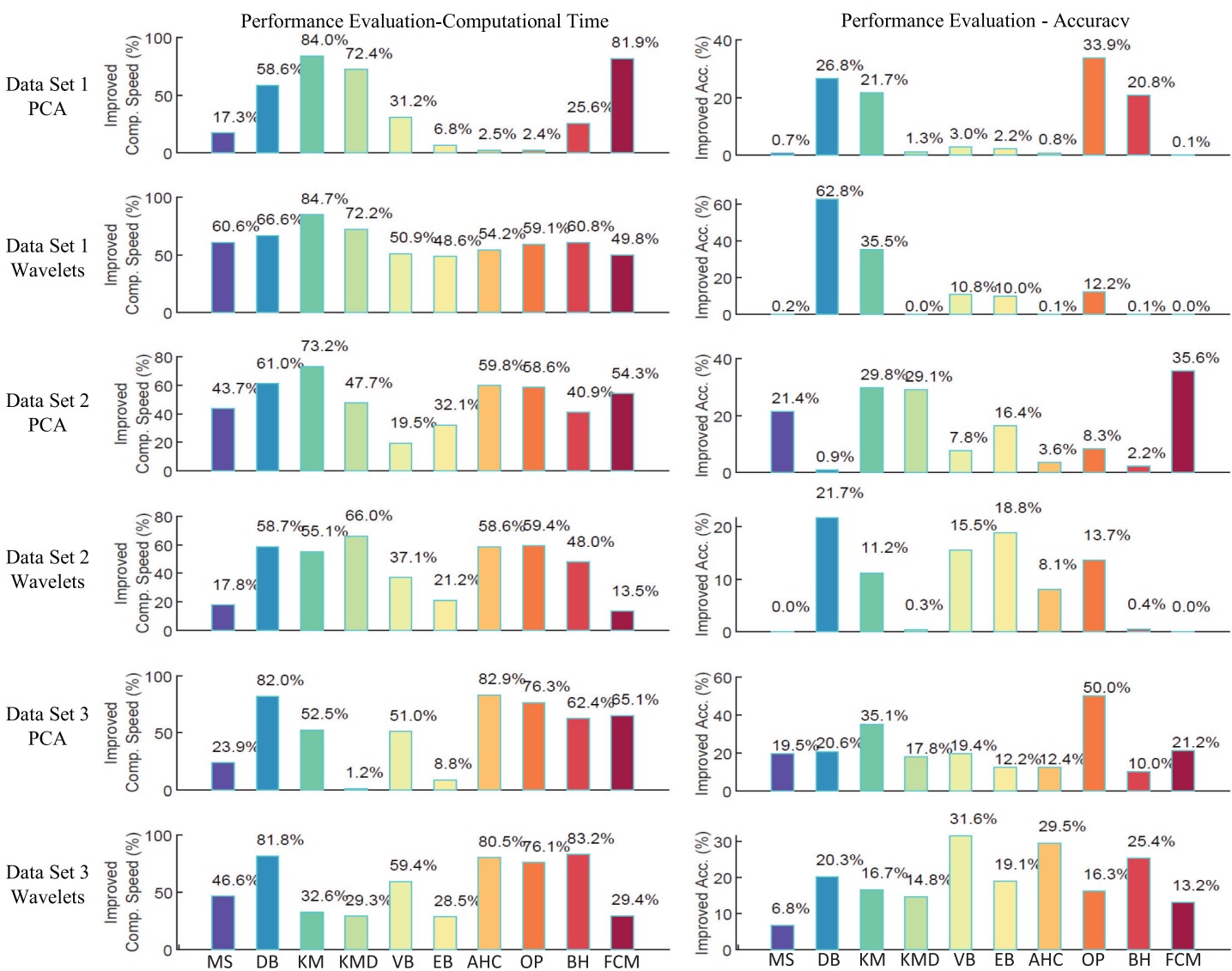

**Fig 7. Illustration of improved computational speed and clustering accuracy.** (a) Improved computational speed in percentage of understudy ten algorithms across six large neural feature sets. (b) Improved clustering accuracy in percentage of understudy ten algorithms across six large neural feature sets. The proposed data-subdivision and unification method has shown a positive trend in improving the performance of spike sorting algorithms. The improvement in reducing computational time is significantly high, while due to maturity of spike of sorting algorithms, accuracy improvement is relatively lower in some of spike sorting algorithms. The average results for 10 repetitive analysis has been presented and it is worth noting that proposed mechanism has shown promising improvement results around all data types and spike sorting algorithms.

and labeled in the ground truth. Labels for three distinguished clusters are provided for each of dataset D1, D2 and D3 in their respective ground truth.

Each spike waveform consists of 64 samples. Haar Wavelets and PCA features are employed to reduce the data dimensionality while preserving the variance of the data and spike information. In case of Haar wavelets transform, optimal wavelet features were selected following the study of Quiroga [79], which implemented the four-level multi resolution decomposition. The 64 wavelet coefficients generated provides unique spike characteristic at different scales and times. As each spike class has different multimodal distribution, the Lilliefors modification of Kolmogorov-Smirnov (KS) test for normality [81] was used to select the optimal wavelet

features. The maximum deviation of multimodal distribution features from normality defines the optimal features. We refer the readers to Quiroga [79] for further explanation. In this context, 10 wavelet features with largest deviation of normality is regarded as optimal wavelet features.

Similarly, 10 PCA features were selected in this study to validate the computational and performance efficiency of the proposed vs conventional algorithms. The PCA components are not scaled to match their explained variances. The individual variances of PCA components are accumulated and the optimal number of PCA components that gives at least 85% of cumulative explained variance are chosen for the analysis. 10 PCA features are required to get at least 85% cumulative explained variance of the 64 dimensional spikes data used in this study.

It is important to note that, the accuracy of clustering algorithms may be affected by the data dimensionality and number of optimal feature sets used. However, in this study the same 10-dimensional features are used for all the algorithms to maintain the consistency while validating the performance outcomes.

The research work is carried out on a personal computer (PC) consisting of Intel (R) Pentium (R) CPU G4560 @3.5GHz, 8 GB of RAM and 64 Bit windows 10 operating system.

## Performance on computational time or speed

To explore and validate the performance of the proposed algorithm, in terms of computational time, as tabulated in Table 2, is estimated using the expression (12).

$$T_s \ (\%) = \ \frac{(C_t - P_t)}{C_t} \times 100 \qquad (12)$$

Where $C_t$ and $P_t$ are computational times of clustering using conventional and proposed algorithms, respectively.

## Performance on clustering accuracy

The clustered spikes from spike sorting algorithms are generally evaluated using the validation indices [46]. In this work clustering accuracy, as is described in [86] and (13), is adopted as a validation index and is calculated using the confusion matrix [87] as in (14).

$$A = \frac{\#\ of\ accurately\ clustered\ spikes}{Total\ \#\ of\ Spikes}\% = \frac{Sum\ of\ Conf.\ Matrix\ Diagonals}{Total\ \#\ of\ Spikes}\% \qquad (13)$$

$$C = \begin{bmatrix} C_{e_1 g_1} & C_{e_1 g_2} \cdots & C_{e_1 g_q} \\ C_{e_2 g_1} & C_{e_2 g_2} \cdots & C_{e_2 g_q} \\ . & . & . \\ . & . & . \\ . & . & . \\ C_{e_m g_1} & C_{e_m g_2} \cdots & C_{e_m g_q} \end{bmatrix} \qquad (14)$$

Where $A$ and $C$ are accuracy index and confusion matrix, respectively. $m$ is total number of estimated clusters and $q$ is total number of clusters in ground truth. $C_{e_i g_i}$ represents the number of spikes estimated and clustered accurately relative to the labels provided with the spikes

**Table 2. Computational times and time based performance improvement for ten clustering algorithms.**

| Algorithm | Method | Computational Time (Seconds) | | | | | |
|---|---|---|---|---|---|---|---|
| | | D1, PCA | D1, WAV | D2, PCA | D2, WAV | D3, PCA | D3, WAV |
| Meanshift | Proposed | 0.3 | 0.02 | 0.43 | 0.13 | 0.11 | 0.03 |
| | Conventional | 0.36 | 0.04 | 0.76 | 0.16 | 0.15 | 0.06 |
| | **Time Saved (%)** | **17.25** | **60.59** | **43.69** | **17.81** | **23.88** | **46.63** |
| DBSCAN | Proposed | 0.75 | 1.6 | 3.26 | 0.54 | 8.76 | 3.84 |
| | Conventional | 1.82 | 4.81 | 8.37 | 1.3 | 48.63 | 21.09 |
| | **Time Saved (%)** | **58.6** | **66.64** | **61.02** | **58.71** | **81.98** | **81.77** |
| Kmeans | Proposed | 0.04 | 0 | 0.01 | 0 | 0.04 | 0.03 |
| | Conventional | 0.28 | 0.03 | 0.03 | 0.01 | 0.09 | 0.05 |
| | **Time Saved (%)** | **84** | **84.7** | **73.2** | **55.14** | **52.55** | **32.59** |
| Kmedoids | Proposed | 0.32 | 0.1 | 0.17 | 0.14 | 1.37 | 1.31 |
| | Conventional | 1.14 | 0.36 | 0.33 | 0.41 | 1.38 | 1.85 |
| | **Time Saved (%)** | **72.39** | **72.2** | **47.67** | **65.99** | **1.19** | **29.26** |
| VBGMM | Proposed | 0.3 | 0.25 | 0.6 | 0.4 | 1.91 | 1.05 |
| | Conventional | 0.44 | 0.51 | 0.75 | 0.63 | 3.9 | 2.57 |
| | **Time Saved (%)** | **31.17** | **50.85** | **19.46** | **37.15** | **51.02** | **59.35** |
| EMGMM | Proposed | 0.43 | 0.32 | 1.2 | 0.46 | 3.39 | 3.07 |
| | Conventional | 0.46 | 0.62 | 1.76 | 0.58 | 3.71 | 4.29 |
| | **Time Saved (%)** | **6.83** | **48.56** | **32.06** | **21.17** | **8.78** | **28.5** |
| Agglomerative | Proposed | 0.18 | 0.07 | 0.06 | 0.06 | 0.2 | 0.2 |
| | Conventional | 0.18 | 0.15 | 0.15 | 0.14 | 1.14 | 1.04 |
| | **Time Saved (%)** | **2.55** | **54.23** | **59.75** | **58.61** | **82.92** | **80.46** |
| OPTICS | Proposed | 1.11 | 0.44 | 0.42 | 0.43 | 1.72 | 1.72 |
| | Conventional | 1.14 | 1.08 | 1.02 | 1.05 | 7.27 | 7.18 |
| | **Time Saved (%)** | **2.35** | **59.15** | **58.56** | **59.37** | **76.34** | **76.07** |
| BIRCH | Proposed | 1.25 | 1.61 | 1.39 | 1.94 | 2.22 | 5.32 |
| | Conventional | 1.68 | 4.11 | 2.35 | 3.73 | 5.9 | 31.56 |
| | **Time Saved (%)** | **25.65** | **60.77** | **40.94** | **47.99** | **62.4** | **83.15** |
| FCM | Proposed | 0.01 | 0.03 | 0.01 | 0.05 | 0.02 | 0.41 |
| | Conventional | 0.08 | 0.05 | 0.02 | 0.06 | 0.05 | 0.59 |
| | **Time Saved (%)** | **81.92** | **49.8** | **54.26** | **13.51** | **65.12** | **29.43** |

data ground truth. Where $e_i$ refers to estimated cluster index and $g_i$ ground truth. The accuracy index highlights the percentage of spikes accurately labelled to the clusters described in the ground truth. There are two scenarios taken into account while calculating accuracies.

$m = q$: when number of clusters estimated are equal to number of clusters in the ground truth. This leads to the square confusion matrix of size $m|_{m=q}$ and the sum of confusion matrix diagonals divided by total number of spikes provides the percentage of accuracy as in Eq (13).

$m \neq q$: when the number of clusters estimated $m$ are not equal to the number of clusters in ground truth $q$, the confusion matrix is generated by taking only the dominant estimated clusters $m$ equal to the total number of clusters $q$ in the ground truth. In case of estimated clusters less than the ground truth clusters, i.e. $m < q$, the confusion matrix is zero padded. The accuracy is calculated by using the expression (13).

The percentage of accuracy enhancement is estimated using the accuracy difference between proposed and conventional methods, which is tabulated in Table 3.

**Table 3. Clustering accuracy and accuracy based performance improvement for ten clustering algorithms.**

| Algorithm | Method | Accuracy (%) | | | | | |
|---|---|---|---|---|---|---|---|
| | | D1, PCA | D1, WAV | D2, PCA | D2, WAV | D3, PCA | D3, WAV |
| Meanshift | Proposed | 89.89 | 97.81 | 83.76 | 94.03 | 72.18 | 81.82 |
| | Conventional | 89.18 | 97.59 | 62.36 | 94 | 52.72 | 75.05 |
| | **Improved Acc. (%)** | **0.71** | **0.23** | **21.4** | **0.03** | **19.46** | **6.76** |
| DBSCAN | Proposed | 87.85 | 92.16 | 34.72 | 84.02 | 72.18 | 72.19 |
| | Conventional | 61.07 | 35.29 | 33.79 | 62.3 | 51.55 | 51.88 |
| | **Improved Acc. (%)** | **26.77** | **56.87** | **0.93** | **21.72** | **20.63** | **20.32** |
| Kmeans | Proposed | 66.35 | 99.38 | 95.21 | 92.78 | 71.63 | 78.87 |
| | Conventional | 44.69 | 63.86 | 65.46 | 81.58 | 36.56 | 62.21 |
| | **Improved Acc. (%)** | **21.66** | **35.52** | **29.76** | **11.19** | **35.06** | **16.66** |
| Kmedoids | Proposed | 98.07 | 99.38 | 77.52 | 93.1 | 61.31 | 83.92 |
| | Conventional | 96.79 | 99.38 | 48.46 | 92.78 | 43.48 | 69.14 |
| | **Improved Acc. (%)** | **1.28** | **0** | **29.06** | **0.32** | **17.82** | **14.78** |
| VBGMM | Proposed | 88.25 | 77.31 | 62.3 | 84.34 | 70.08 | 63.51 |
| | Conventional | 85.26 | 66.47 | 54.52 | 68.85 | 50.66 | 31.93 |
| | **Improved Acc. (%)** | **2.98** | **10.85** | **7.77** | **15.49** | **19.42** | **31.58** |
| EMGMM | Proposed | 91.31 | 90.37 | 72.97 | 84.66 | 74.15 | 57.21 |
| | Conventional | 89.15 | 80.41 | 56.61 | 65.84 | 61.95 | 38.15 |
| | **Improved Acc. (%)** | **2.16** | **9.97** | **16.36** | **18.82** | **12.2** | **19.05** |
| Agglomerative | Proposed | 94.55 | 99.26 | 88.46 | 96 | 56.15 | 80.88 |
| | Conventional | 93.7 | 99.21 | 84.86 | 87.91 | 43.75 | 51.4 |
| | **Improved Acc. (%)** | **0.85** | **0.06** | **3.6** | **8.09** | **12.4** | **29.48** |
| OPTICS | Proposed | 76.58 | 29.42 | 31 | 26.04 | 62.33 | 20.58 |
| | Conventional | 42.67 | 17.18 | 22.71 | 12.38 | 12.34 | 4.28 |
| | **Improved Acc. (%)** | **33.9** | **12.24** | **8.29** | **13.66** | **49.99** | **16.29** |
| BIRCH | Proposed | 93.19 | 99.26 | 86.83 | 93.33 | 54.73 | 80.88 |
| | Conventional | 72.43 | 99.18 | 84.66 | 92.89 | 44.76 | 55.53 |
| | **Improved Acc. (%)** | **20.76** | **0.09** | **2.18** | **0.44** | **9.96** | **25.35** |
| FCM | Proposed | 71.78 | 99.38 | 84.98 | 92.63 | 60.99 | 77.15 |
| | Conventional | 71.72 | 99.38 | 49.42 | 92.6 | 39.77 | 63.97 |
| | **Improved Acc. (%)** | **0.06** | **0** | **35.56** | **0.03** | **21.22** | **13.18** |

## Clustering results

To highlight enhancement in clustering quality, visual representation of clusters estimated using proposed and conventional methods employing OPTICS on dataset 3 with PCA features and DBSCAN on dataset 1 with Wavelet features, gives 49.99 and 56.87 percent accuracy improvement in the clustering results with respect to the ground truth as in Table 3. The illustration of clustering results for aforementioned examples is shown in Figs 8 and 9 respectively. It is clear from the results that proposed methodology generates significantly superior results in contrast to conventional methods.

## Discussion

It is largely observed from the results and performance evaluation that the proposed algorithm shows continuous improvement around all algorithms and datasets. The accuracy is improved up to 56.87% while computational time is reduced up to 84.7%. Hence, proposed mechanism has significant impact on enhancing the speed and accuracy of the spike sorting process. In

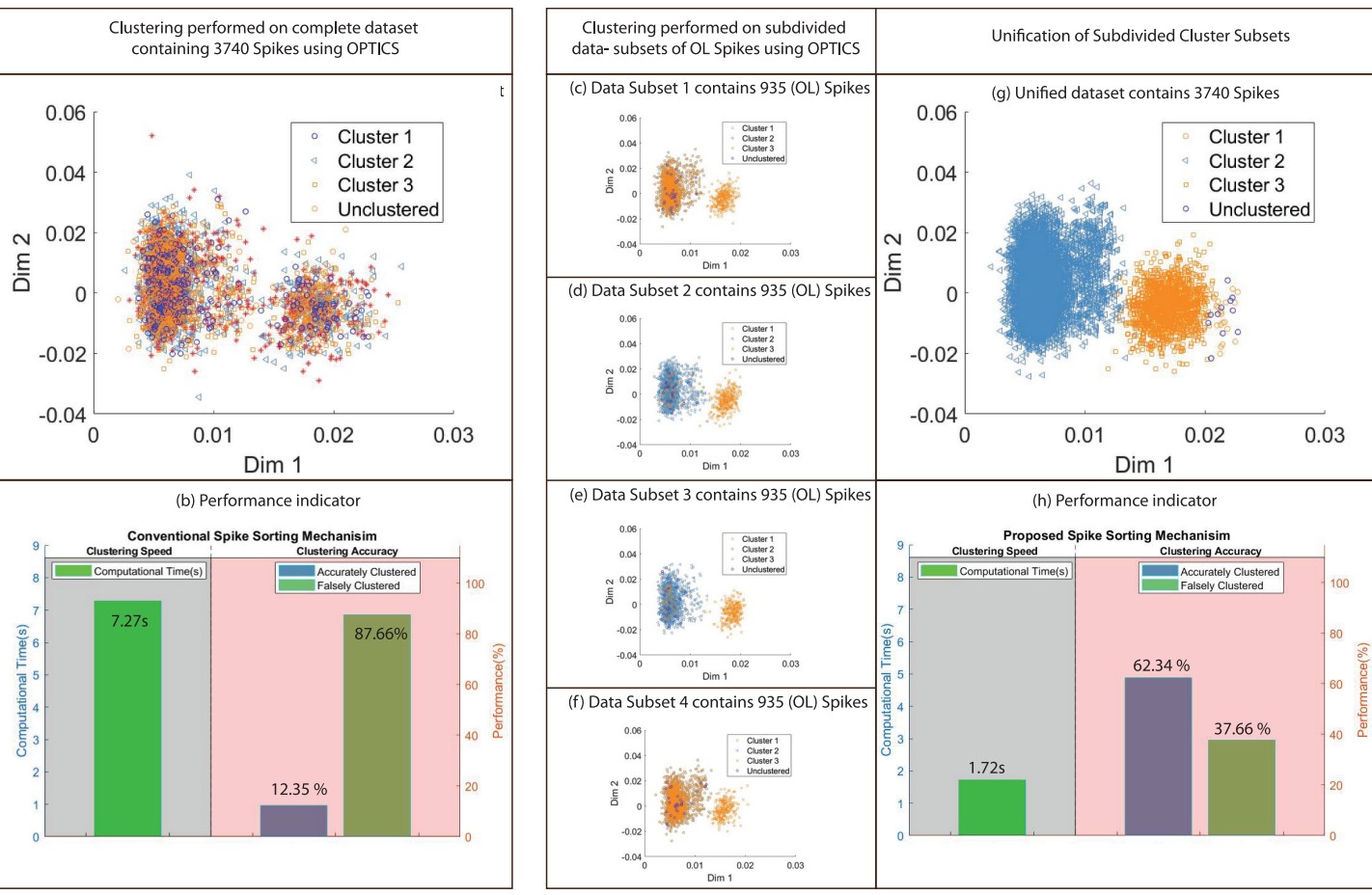

**Fig 8. Comparison of clustering results obtained using conventional and proposed mechanism employing OPTICS with dataset 3 and PCA features.** (a) Clustering results using conventional spike sorting method applied on complete dataset containing 3740 Spikes. (b) Performance indication of clustering results based on computational time/speed and clustering accuracy. (c)-(f) Clustering results using proposed spike sorting mechanism applied on data-subdivision of optimal length i.e 935 for OPTICS. (g) Unification of subdivided cluster subsets. (h) Performance indication of clustering results using proposed spike sorting method.

term of clustering accuracy, DBSCAN demonstrates high accuracy improvement of 56.87 percent followed by OPTICS at 49.99 percent. In terms of computational time, Kmeans shows highest computational speed enhancement of 84.7 percent followed by BIRCH with computational speed enhancement of 83.15 percent. In terms of parameter tuning complexity, Mean-Shift, FCM and Gaussian Mixture models require one parameter to tune, DBSCAN and OPTICS require two and BIRCH requires three parameters to tune to perform their operations. All the supervised clustering algorithms including Kmeans, Kmedoids and Agglomerative require single parameter to tune. In terms of robustness, Kmeans, Kmedoids,FCM gives different results at every iteration, however, Meanshift, EMGMM, VMGMM, Agglomerative, DBSCAN, OPTICS, BIRCH converged to same results after each iteration. For simplicity of the presentation, the presented results are averaged over 10 repetitions.

## Software implementation

The software for proposed mechanism is implemented using MATLAB as shown in Fig 10. The free access to open source software for academic purpose is provided with detailed user

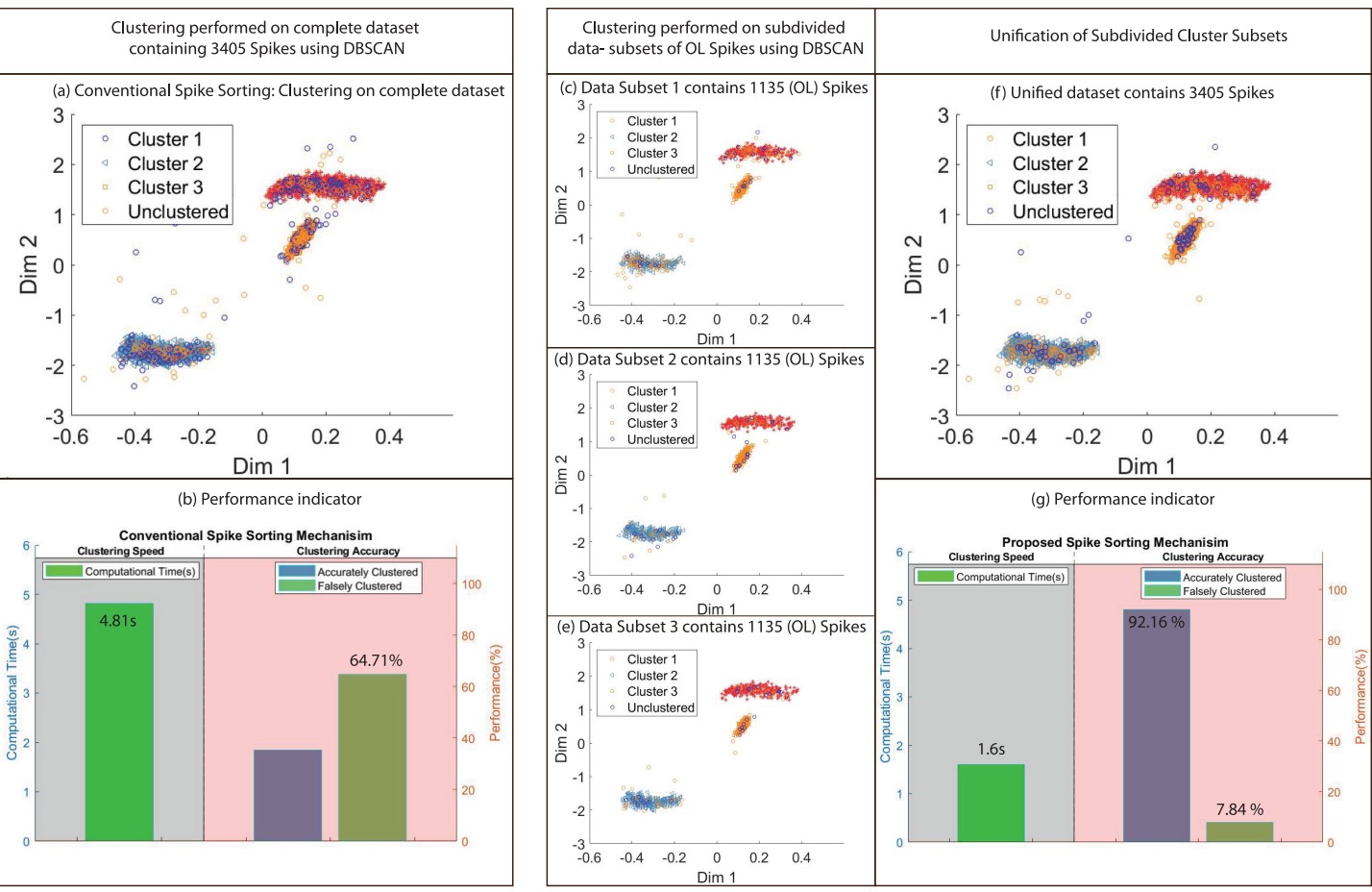

**Fig 9. Comparison of clustering results obtained using conventional and proposed mechanism employing DBSCAN with dataset 1 and Wavelet features.** (a) Clustering results using conventional spike sorting method applied on complete dataset containing 3405 Spikes. (b) Performance indication of clustering results based on computational time/speed and clustering accuracy. (c)-(f) Clustering results using proposed spike sorting mechanism applied on data-subdivision of optimal length i.e 1135 for DBSCAN. (g) Unification of subdivided cluster subsets. (h) Performance indication of clustering results using proposed spike sorting method.

instructions online at: https://github.com/ermasood/Handling-Larger-Data-Sets-for-Clustering. The software yields the clustering labels with high accuracy and in a fast and efficient way. The first graph in the software window shows the clustered spikes and the second graph illustrates the clustered features of the inputted data. MATLAB codes provided are tested on 2019b and 2018b MATLAB versions. Additionally,'Linspecer.m' file [88] from MathWorks is required to generate attractive colour combinations and shades for beautiful visualisations.

## Conclusion

Neural spike sorting is prerequisite to deciphering useful information from electrophysiological data recorded from the brain, in vitro and/or in vivo. Significant advancements in nanotechnology and nano fabrication has enabled neuroscientists and engineers to capture the electrophysiological activities of the brain at very high resolution, data rate and fidelity. However, the evolution in spike sorting algorithms to deal with the aforementioned technological advancement and capability to quantify higher density data sets is somewhat limited. It is observed from the experiments that larger datasets highly effect the computational time

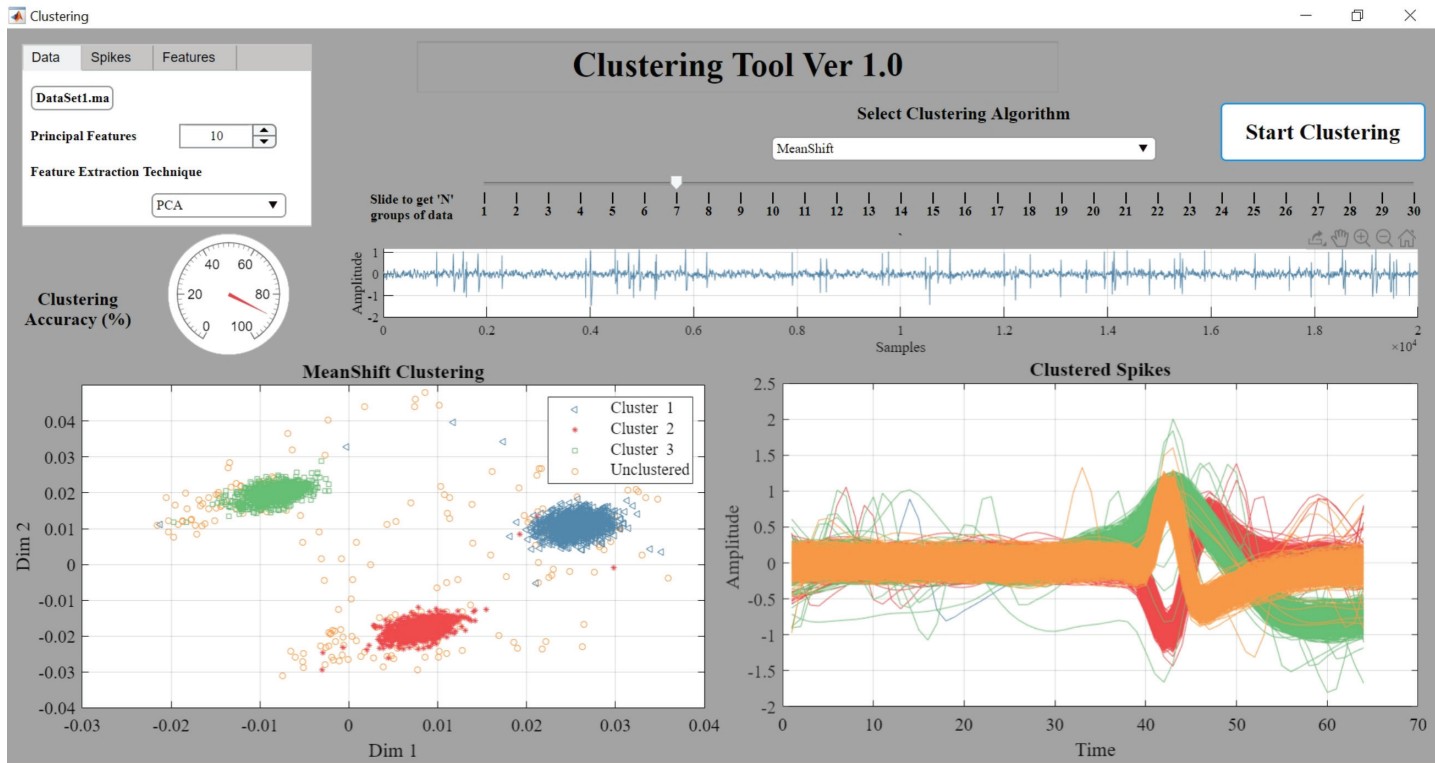

**Fig 10. Software for proposed clustering mechanism.**

required to perform clustering. To address this challenge, a novel clustering mechanism is proposed to handle large datasets efficiently and with higher accuracy. The proposed mechanism resolves the issue of high computational time and reduced accuracy in conventional method. The proposed algorithms has demonstrated up to 84% and 56% improvement in terms of computational time and clustering accuracy, respectively. The proposed framework is validated by applying on ten widely used clustering algorithms and six large data sets. PCA and Haar wavelets features are employed for consistency during the clustering process. A MATLAB software of the proposed mechanism is also developed and provided to assist the researchers, active in this domain.

## Supporting information

**S1 Data.**
(ZIP)

## Author Contributions

**Conceptualization:** Masood Ul Hassan, Rakesh Veerabhadrappa, Asim Bhatti.

**Data curation:** Masood Ul Hassan, Rakesh Veerabhadrappa, Asim Bhatti.

**Formal analysis:** Masood Ul Hassan, Rakesh Veerabhadrappa, Asim Bhatti.

**Funding acquisition:** Masood Ul Hassan, Rakesh Veerabhadrappa, Asim Bhatti.

**Investigation:** Masood Ul Hassan, Rakesh Veerabhadrappa, Asim Bhatti.

**Methodology:** Masood Ul Hassan, Rakesh Veerabhadrappa, Asim Bhatti.

**Project administration:** Masood Ul Hassan, Rakesh Veerabhadrappa, Asim Bhatti.

**Resources:** Masood Ul Hassan, Rakesh Veerabhadrappa, Asim Bhatti.

**Software:** Masood Ul Hassan, Rakesh Veerabhadrappa, Asim Bhatti.

**Supervision:** Masood Ul Hassan, Rakesh Veerabhadrappa, Asim Bhatti.

**Validation:** Masood Ul Hassan, Rakesh Veerabhadrappa, Asim Bhatti.

**Visualization:** Masood Ul Hassan, Rakesh Veerabhadrappa, Asim Bhatti.

**Writing – original draft:** Masood Ul Hassan, Rakesh Veerabhadrappa, Asim Bhatti.

**Writing – review & editing:** Masood Ul Hassan, Rakesh Veerabhadrappa, Asim Bhatti.

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
