## [Decision Letter · Decision Letter 0]

15 Oct 2019

PONE-D-19-23638

Efficient Neural Spike Sorting using Data Subdivision and Unification

PLOS ONE

Dear Dr. Bhatti,

Thank you for submitting your manuscript to PLOS ONE. After careful consideration, we feel that it has merit but does not fully meet PLOS ONE’s publication criteria as it currently stands. Therefore, we invite you to submit a revised version of the manuscript that addresses the points raised during the review process.

(1) Abstract.

What do you mean by "big data dynamics"? This term is ambiguous. Please change this sentence and the next one, immediately following. Rephrase in order to avoid reference to other algorithms reported in the literature if the Authors do not cite explicitly which ones. The current reference is too general and inappropriate.

How many datasets did you use?

(2) Introduction.

Please, focus the introduction on the problems addressed and thoroughly review the literature and the current state of the art in the field.

1. The review of the literature is not complete, because it missed one key important paper related to this topic, in particular because that paper has introduced for the first time a series of steps that are very close, if not identical, to the steps of data subdivision, clusters formed for each sub-set, unification process by merging neighbor clusters in feature space, thus achieving unified clusters in the end. This paper is the following:

- Aksenova TI, Chibirova OK, Dryga OA, Tetko IV, Benabid AL, Villa AE. An unsupervised automatic method for sorting neuronal spike waveforms in awake and freely moving animals. Methods. 2003; 30(2):178-187. doi: 10.1016/S1046-2023(03)00079-3 : this is the very first paper (2003) to describe unsupervised neural spike sorting based on a fast implementation suitable for real-time application for high-density neural probes.

With respect to application of spike sorting to online experimental procedures, the Authors should also mention:

- Abeles M, Goldstein MH. Multispike train analysis. Proceedings of the IEEE. 1977; 65(5):762-773. doi:10.1109/PROC.1977.10559 : this is a seminal paper (1977) for detecting and identifying the spikes in multispike trains based on signal detection by template matching.

- Wouters J, Kloosterman F, Bertrand A. Towards online spike sorting for high-density neural probes using discriminative template matching with suppression of interfering spikes. J Neural Eng. 2018; 15(5):056005. doi: 10.1088/1741-2552/aace8a : a fast and computationally cheap method for real-time applications.

Consider  recently developed spike sorting algorithms :

Chung, Jason E., Jeremy F. Magland, Alex H. Barnett, Vanessa M. Tolosa, Angela C. Tooker, Kye Y. Lee, Kedar G. Shah, Sarah H. Felix, Loren M. Frank, and Leslie F. Greengard. "A fully automated approach to spike sorting." Neuron 95, no. 6 (2017): 1381-1394.

A more satisfactory review of the literature should also include:

- Zamani M, Demosthenous A. (2014) Feature extraction using extrema sampling of discrete derivatives for spike sorting in implantable upper-limb neural prostheses. IEEE Trans Neural Syst Rehabil Eng. 2014 Jul;22(4):716-726. doi: 10.1109/TNSRE.2014.2309678.

(3) Materials and Methods.

The Authors mention several times the problem of noisy recordings, but they do not examine which types of noise --and/or artifacts-- are present and the methods to face this problem that have been described in the recent literature.

A better way to compare the methods presented by the Authors in their Table 2 and Table 3 could have been to add several known levels of noise to the same benchmarked data set and see how performances and accuracies allow to discriminate the most robust algorithms.

To this end, the Authors should consider these papers:

- Choi JH, Jung HK, Kim T. (2006) A new action potential detector using the MTEO and its effects on spike sorting systems at low signal-to-noise ratios. IEEE Trans Biomed Eng. 2006 Apr;53(4):738-46. doi: 10.1109/TBME.2006.870239

- Paralikar KJ, Rao CR, Clement RS. (2009) New approaches to eliminating common-noise artifacts in recordings from intracortical microelectrode arrays: inter-electrode correlation and virtual referencing. J Neurosci Methods. 2009 Jun 30;181(1):27-35. doi: 10.1016/j.jneumeth.2009.04.014.

- Pillow JW1, Shlens J, Chichilnisky EJ, Simoncelli EP. (2013) A model-based spike sorting algorithm for removing correlation artifacts in multi-neuron recordings. PLoS One. 2013 May 3;8(5):e62123. doi: 10.1371/journal.pone.0062123.

- Takekawa T, Ota K, Murayama M, Fukai T. (2014) Spike detection from noisy neural data in linear-probe recordings. Eur J Neurosci. 2014 Jun;39(11):1943-50. doi: 10.1111/ejn.12614:  an older reference to Takekawa is provided but it should be replaced by this one .

The Authors discuss Spike sorting accuracy (Subsection 3.5) but false alarm ratio is also an extremely important feature to be considered (and discussed in several papers cited above) for the evaluation of the quality of neural spike sorting.

(4) Results.

The Authors should provide the MATLAB codes, with the description of the MATLAB version and environment, of their algorithms.  They compare many methods developed elsewhere and it is of paramount importance to assess that the Authors' implementation follows exactly the algorithms cited in the literature.

A test against a surrogate data set could also be informative for the readers to be convinced of their superior efficiency in the spike sorting procedure claimed by the Authors.

-Optimal length: describe how relevant it is to have the 'optimal length'.

Please, substantiate: 'OL parameter is dependent on the algorithm type rather than on the data dynamics.' The spiking rates may vary by 2 orders of magnitude, so you may end up with clusters that simply don't have enough spikes?

Clarify labeling in Figure 4.

Unification of subclusters:

Describe in detail how you account for differing variances in different dimensions (i.e. principal components).   Explain what 'the standard distribution and normal distribution curves' are.

In general, describe how this technique is applied to the data. Do you apply it to sequential segments, blocks of segments or pairwise across the recording?

Performance evaluation:

Why do you choose two examples where both the conventional and your method do not work for showing performance improvement?

Figure 6: Why those spline fits? Suggests that the different methods are related, please, explain.

Table 3: Numbers suggest a very high accuracy, and no error estimate is given. How did you achieve such a high precision? K-means for example is known to give very different results in different runs. Are these averages over multiple runs? And does the K-means example involve multiple runs to obtain stable clusters? Which of these algorithms converge to the same result every time they are run? Could part of your accuracy improvement be due to running K-means more often, effectively averaging results?

Figure 7:

Lines/symbols are overlapping to an extent that this figure becomes uninformative. Maybe separate plots or cluster centroids for different segments? Please, provide a plot showing the temporal stationarity of firing rates (for different segments).

Please, clarify description of the algorithm concerning temporal speedup. What is the advantage of independent clustering? How does your method compare to a density based approach?

clustering accuracy:

The measure you are using puts a higher weight on large clusters with a lot of spikes. In many datasets, these are multiunit clusters that are hard to separate. It would be nice to have some measure of temporal stationarity.

We would appreciate receiving your revised manuscript by Nov 29 2019 11:59PM. To enhance the reproducibility of your results, we recommend that if applicable you deposit your laboratory protocols in protocols.io, where a protocol can be assigned its own identifier (DOI) such that it can be cited independently in the future. For instructions see: http://journals.plos.org/plosone/s/submission-guidelines#loc-laboratory-protocols

We look forward to receiving your revised manuscript.

Kind regards,

Gennady Cymbalyuk, Ph.D.

Academic Editor

PLOS ONE

Journal Requirements:

2)  Thank you for stating the following in the Acknowledgments Section of your manuscript:

[The research work is fully supported by Neural and Cognitive Systems Lab at Institute

for Intelligent Systems Research and Innovation, Deakin University.]

 [The author(s) received no specific funding for this work.]

Please include the updated Funding Statement in your cover letter. We will change the online submission form on your behalf.

Reviewers' comments:

Reviewer's Responses to Questions

**Comments to the Author**

1. Is the manuscript technically sound, and do the data support the conclusions?

Reviewer #1: No

Reviewer #2: Partly

2. Has the statistical analysis been performed appropriately and rigorously? 

Reviewer #1: No

Reviewer #2: N/A

3. Have the authors made all data underlying the findings in their manuscript fully available?

Reviewer #1: Yes

Reviewer #2: No

4. Is the manuscript presented in an intelligible fashion and written in standard English?

Reviewer #1: No

Reviewer #2: Yes

5. Review Comments to the Author

Reviewer #1: The authors address a relevant problem in spike sorting, namely how to deal with datasets from recordings that become increasingly long due to technological advances in recording techniques.

However, the presentation of their results and statistical analyses do not allow me to make any judgements about the validity of their approach. In fact, I believe that the amount of changes that would be necessary for a revised version of this manuscript would effectively amount to a resubmission of the manuscript.

Specifically, the introduction consists of a rather broad discussion about measuring brain activity and its relevance (not immediately related to the manuscript), but almost completely ignores recently developed spike sorting algorithms (e.g.

Chung, Jason E., Jeremy F. Magland, Alex H. Barnett, Vanessa M. Tolosa, Angela C. Tooker, Kye Y. Lee, Kedar G. Shah, Sarah H. Felix, Loren M. Frank, and Leslie F. Greengard. "A fully automated approach to spike sorting." Neuron 95, no. 6 (2017): 1381-1394.

and the sorting algorithms they use for comparison). It would good to have an introduction more specific to the manuscript and especially describing the current state of the art in the field.

-Optimal length: what I missed here is a discussion of how relevant it is to have the 'optimal length'. Can I be off by a factor of 2 and it doesn't really matter?

Also, I'm not sure how the authors come up with this claim: 'OL parameter is dependent on the algorithm type rather than on the data dynamics.' This may be the case in machine learning examples, but here spiking rates may vary by 2 orders of magnitude, so you may end up with clusters that simply don't have enough spikes?

Labelling in Figure 4 is messy, I don't understand what is plotted.

Unification of subclusters:

I don't understand how you account for differing variances in different dimensions (i.e. principal components). And for distances, in 1D, the 95% claim is valid, but here you're talking about volumes. And I'm completely lost about what 'the standard distribution and normal distribution curves' are.

In general, I'm wondering how this technique is applied to the data. Do you apply it to sequential segments, blocks of segments or pairwise across the recording?

Performance evaluation:

Most strikingly, why do you choose two examples where both the conventional and your method do not work for showing performance improvement? Seems not relevant to the reader.

Figure 6: Why those spline fits? Suggests that the different methods are related, but I do not see how.

Table 3: Numbers suggest a very high accuracy, and no error estimate is given. How did you achieve such a high precision? K-means for example is known to give very different results in different runs. Are these averages over multiple runs? And does the K-means example involve multiple runs to obtain stable clusters? Which of these algorithms converge to the same result every time they are run? I am also concerned that part of your accuracy improvement might be due to running K-means more often, effectively averaging results.

Figure 7:

Lines/symbols are overlapping to an extent that this figure becomes uninformative. Maybe separate plots or cluster centroids for different segments? What I am missing here is also a plot showing the temporal stationarity of firing rates (for different segments).

temporal speedup:

If I understood things correctly (and I'm not sure I did), PCA/Wavelet is run on the whole dataset to obtain low dimensional representations of spikes. Then batches of N spikes are clustered. That sounds similar to what Kilosort does, except that batches are used for optimizing clusters rather than clustering them independently. What is the advantage of independent clustering? Mountainsort on the other hand follows a density based approach, which also seems to scale quite well with recording size. How does your method compare to a density based approach?

clustering accuracy:

The measure you are using puts a higher weight on large clusters with a lot of spikes. In many datasets, these are multiunit clusters that are hard to separate. Also, It would be nice to have some measure of temporal stationarity.

Abstract:

6 or 3 datasets?

Reviewer #2: GENERAL COMMENTS

The Authors present an interesting manuscript about an efficient method to apply spike sorting on large data sets -- in the order of several hundreds of multiple spike trains recorded simultaneously. This topic is central for any project aimed at real-time decoding of brain activity, in particular for brain-machine interfaces. The paper is well written and reads easily. The main principles and methods are clearly presented and the figures are well done. The recommendation is to accept the paper, but there are few suggested corrections to introduce and the paper may be accepted only after appropriate amendments are introduced.

SPECIFIC COMMENTS

(1) Abstract.

The Authors use the expression "big data dynamics". What does it mean? This sounds a bit weird because it may assume so many different meanings. Please change this sentence and the next one, immediately following. Rephrase in order to avoid reference to other algorithms reported in the literature if the Authors do not cite explicitely which ones. The current reference is too general and inappropriate.

(2) Introduction.

1. The review of the literature is not complete, because it missed one key important paper related to this topic, in particular because that paper has introduced for the first time a series of steps that are very close, if not identical, to the steps of data subdivision, clusters formed for each sub-set, unification process by merging neighbor clusters in feature space, thus achieving unified clusters in the end. This paper is the following:

- Aksenova TI, Chibirova OK, Dryga OA, Tetko IV, Benabid AL, Villa AE. An unsupervised automatic method for sorting neuronal spike waveforms in awake and freely moving animals. Methods. 2003; 30(2):178-187. doi: 10.1016/S1046-2023(03)00079-3 : this is the very first paper (2003) to describe unsupervised neural spike sorting based on a fast implementation suitable for real-time application for high-density neural probes.

With respect to application of spike sorting to online experimental procedures, the Authors should also mention:

- Abeles M, Goldstein MH. Multispike train analysis. Proceedings of the IEEE. 1977; 65(5):762-773. doi:10.1109/PROC.1977.10559 : this is a seminal paper (1977) for detecting and identifying the spikes in multispike trains based on signal detection by template matching.

- Wouters J, Kloosterman F, Bertrand A. Towards online spike sorting for high-density neural probes using discriminative template matching with suppression of interfering spikes. J Neural Eng. 2018; 15(5):056005. doi: 10.1088/1741-2552/aace8a : a fast and computationally cheap method for real-time applications.

A more satisfactory review of the literature should also include:

- Zamani M, Demosthenous A. (2014) Feature extraction using extrema sampling of discrete derivatives for spike sorting in implantable upper-limb neural prostheses. IEEE Trans Neural Syst Rehabil Eng. 2014 Jul;22(4):716-726. doi: 10.1109/TNSRE.2014.2309678.

(3) Materials and Methods.

The Authors mention several times the problem of noisy recordings, but they do not examine which types of noise --and/or artifacts-- are present and the methods to face this problem that have been described in the recent literature.

A better way to compare the methods presented by the Authors in their Table 2 and Table 3 could have been to add several known levels of noise to the same benchmarked data set and see how performances and accuracies allow to discriminate the most robust algorithms.

To this end, the Authors should consider these papers:

- Choi JH, Jung HK, Kim T. (2006) A new action potential detector using the MTEO and its effects on spike sorting systems at low signal-to-noise ratios. IEEE Trans Biomed Eng. 2006 Apr;53(4):738-46. doi: 10.1109/TBME.2006.870239

- Paralikar KJ, Rao CR, Clement RS. (2009) New approaches to eliminating common-noise artifacts in recordings from intracortical microelectrode arrays: inter-electrode correlation and virtual referencing. J Neurosci Methods. 2009 Jun 30;181(1):27-35. doi: 10.1016/j.jneumeth.2009.04.014.

- Pillow JW1, Shlens J, Chichilnisky EJ, Simoncelli EP. (2013) A model-based spike sorting algorithm for removing correlation artifacts in multi-neuron recordings. PLoS One. 2013 May 3;8(5):e62123. doi: 10.1371/journal.pone.0062123.

- Takekawa T, Ota K, Murayama M, Fukai T. (2014) Spike detection from noisy neural data in linear-probe recordings. Eur J Neurosci. 2014 Jun;39(11):1943-50. doi: 10.1111/ejn.12614: an older reference to Takekawa is provided but it should be replaced by this one .

The Authors discuss Spike sorting accuracy (Subsection 3.5) but false alarm ratio is also an extremely important feature to be considered (and discussed in several papers cited above) for the evaluation of the quality of neural spike sorting.

(4) Results.

The Authors should provide the MATLAB codes, with the description of the MATLAB version and environment, of their algorithms. They compare many methods developed elsewhere and it is of paramount importance to assess that the Authors' implementation follows exactly the algorithms cited in the literature.

A test against a surrogate data set could also be informative for the readers to be convinced of their superior efficiency in the spike sorting procedure claimed by the Authors.

6. PLOS authors have the option to publish the peer review history of their article (what does this mean?). If published, this will include your full peer review and any attached files.

Reviewer #1: No

Reviewer #2: No

---

## [Author Response · Author response to Decision Letter 0]

21 Nov 2019

Below are the suggested revisions according to valuable comments from the reviewers.

1) Abstract.

1. What do you mean by "big data dynamics"? This term is ambiguous. Please change this sentence and the next one, immediately following. Rephrase in order to avoid reference to other algorithms reported in the literature if the Authors do not cite explicitly which ones. The current reference is too general and inappropriate.

Author Response: The manuscript has been updated according to reviewers comment (Line 16 to 20).

2. How many datasets did you use? 

Author Response: The manuscript has been updated according to reviewers comment (Line 24 to 26).

2) Introduction.

Please, focus the introduction on the problems addressed and thoroughly review the literature and the current state of the art in the field.

1. The review of the literature is not complete, because it missed one key important paper related to this topic, in particular because that paper has introduced for the first time a series of steps that are very close, if not identical, to the steps of data subdivision, clusters formed for each sub-set, unification process by merging neighbor clusters in feature space, thus achieving unified clusters in the end. This paper is the following:

Aksenova TI, Chibirova OK, Dryga OA, Tetko IV, Benabid AL, Villa AE. An unsupervised automatic method for sorting neuronal spike waveforms in awake and freely moving animals. Methods. 2003; 30(2):178-187. doi: 10.1016/S1046-2023(03)00079-3 : this is the very first paper (2003) to describe unsupervised neural spike sorting based on a fast implementation suitable for real-time application for high-density neural probes.

Author Response: The manuscript has been updated according to reviewers comment (Line 232 to 234).

2. With respect to application of spike sorting to online experimental procedures, the Authors should also mention:

a) Abeles M, Goldstein MH. Multispike train analysis. Proceedings of the IEEE. 1977; 65(5):762-773. doi:10.1109/PROC.1977.10559 : this is a seminal paper (1977) for detecting and identifying the spikes in multispike trains based on signal detection by template matching.

Author Response: The manuscript has been updated according to reviewers comment (Line 87 to 90).

b) Wouters J, Kloosterman F, Bertrand A. Towards online spike sorting for high-density neural probes using discriminative template matching with suppression of interfering spikes. J Neural Eng. 2018; 15(5):056005. doi: 10.1088/1741-2552/aace8a : a fast and computationally cheap method for real-time applications.

Author Response: The manuscript has been updated according to reviewers comment (Line 109 to 111).

3. Consider recently developed spike sorting algorithms :

Chung, Jason E., Jeremy F. Magland, Alex H. Barnett, Vanessa M. Tolosa, Angela C. Tooker, Kye Y. Lee, Kedar G. Shah, Sarah H. Felix, Loren M. Frank, and Leslie F. Greengard. "A fully automated approach to spike sorting." Neuron 95, no. 6 (2017): 1381-1394.

Author Response: The manuscript has been updated according to reviewers comment (Line 123 to 127).

4. A more satisfactory review of the literature should also include:

Zamani M, Demosthenous A. (2014) Feature extraction using extrema sampling of discrete derivatives for spike sorting in implantable upper-limb neural prostheses. IEEE Trans Neural Syst Rehabil Eng. 2014 Jul;22(4):716-726. doi: 10.1109/TNSRE.2014.2309678.

Author Response: The manuscript has been updated according to reviewers comment (Line 95 to 98).

3) Materials and Methods.

The Authors mention several times the problem of noisy recordings, but they do not examine which types of noise --and/or artifacts-- are present and the methods to face this problem that have been described in the recent literature. A better way to compare the methods presented by the Authors in their Table 2 and Table 3 could have been to add several known levels of noise to the same benchmarked data set and see how performances and accuracies allow to discriminate the most robust algorithms.

1. To this end, the Authors should consider these papers:

Choi JH, Jung HK, Kim T. (2006) A new action potential detector using the MTEO and its effects on spike sorting systems at low signal-to-noise ratios. IEEE Trans Biomed Eng. 2006 Apr;53(4):738-46. doi: 10.1109/TBME.2006.870239

Paralikar KJ, Rao CR, Clement RS. (2009) New approaches to eliminating common-noise artifacts in recordings from intracortical microelectrode arrays: inter-electrode correlation and virtual referencing. J Neurosci Methods. 2009 Jun 30;181(1):27-35. doi: 10.1016/j.jneumeth.2009.04.014.

Pillow JW1, Shlens J, Chichilnisky EJ, Simoncelli EP. (2013) A model-based spike sorting algorithm for removing correlation artifacts in multi-neuron recordings. PLoS One. 2013 May 3;8(5):e62123. doi: 10.1371/journal.pone.0062123.

Takekawa T, Ota K, Murayama M, Fukai T. (2014) Spike detection from noisy neural data in linear-probe recordings. Eur J Neurosci. 2014 Jun;39(11):1943-50. doi: 10.1111/ejn.12614: an older reference to Takekawa is provided but it should be replaced by this one .

Author Response: The manuscript has been updated according to reviewers comment (Line 74 to 87).

2. The Authors discuss Spike sorting accuracy (Subsection 3.5) but false alarm ratio is also an extremely important feature to be considered (and discussed in several papers cited above) for the evaluation of the quality of neural spike sorting.

Author Response: The manuscript has been updated according to reviewers comment (Line 270 to 286).

4) Results.

1. The Authors should provide the MATLAB codes, with the description of the MATLAB version and environment, of their algorithms. They compare many methods developed elsewhere and it is of paramount importance to assess that the Authors' implementation follows exactly the algorithms cited in the literature.

Author Response: The manuscript has been updated according to reviewers comment (Line 316 to 324).

2. A test against a surrogate data set could also be informative for the readers to be convinced of their superior efficiency in the spike sorting procedure claimed by the Authors.

Author Response: The manuscript has been updated according to reviewers comment (Line 250 to 256).

3. Optimal length: describe how relevant it is to have the 'optimal length'. What I missed here is a discussion of how relevant it is to have the 'optimal length'. Can I be off by a factor of 2 and it doesn't really matter?

Author Response: The manuscript has been updated according to reviewers comment (Line 205 to 209).

4. Please, substantiate 'OL parameter is dependent on the algorithm type rather than on the data dynamics.' The spiking rates may vary by 2 orders of magnitude, so you may end up with clusters that simply don't have enough spikes?

Author Response: The manuscript has been updated according to reviewers comment (Line 228 to 231).

5. Clarify labeling in Figure 4., Labelling in Figure 4 is messy, I don't understand what is plotted.

Author Response: The figure has been updated according to reviewers comment (Figure 4).

5) Unification of subclusters:

Describe in detail how you account for differing variances in different dimensions (i.e. principal components). Explain what 'the standard distribution and normal distribution curves' are. In general, describe how this technique is applied to the data. Do you apply it to sequential segments, blocks of segments or pairwise across the recording?

I don't understand how you account for differing variances in different dimensions (i.e. principal components). And for distances, in 1D, the 95% claim is valid, but here you're talking about volumes. And I'm completely lost about what 'the standard distribution and normal distribution curves' are.

In general, I'm wondering how this technique is applied to the data. Do you apply it to sequential segments, blocks of segments or pairwise across the recording?

Author Response: The manuscript has been updated according to reviewers comment (Line 220 to 228).

6) Performance evaluation: Why do you choose two examples where both the conventional and your method do not work for showing performance improvement?

Author Response: The manuscript has been updated according to reviewers comment (Line 288 to 295).

7) Figure 6: Why those spline fits? Suggests that the different methods are related, please, explain.

Author Response: The figure has been updated according to reviewers comment (Figure 6).

8) Table 3: Numbers suggest a very high accuracy, and no error estimate is given. How did you achieve such a high precision? K-means for example is known to give very different results in different runs. Are these averages over multiple runs? And does the K-means example involve multiple runs to obtain stable clusters? Which of these algorithms converge to the same result every time they are run? Could part of your accuracy improvement be due to running K-means more often, effectively averaging results? 

Author Response: The manuscript has been updated according to reviewers comment (Line 309 to 313).

9) Figure 7: Lines/symbols are overlapping to an extent that this figure becomes uninformative. Maybe separate plots or cluster centroids for different segments? Please, provide a plot showing the temporal stationarity of firing rates (for different segments). 

Author Response: The figures has been updated according to reviewer’s comment (Figure 7 and Figure 8).

10) Temporal speedup: Please, clarify description of the algorithm concerning temporal speedup. What is the advantage of independent clustering? How does your method compare to a density based approach?

If I understood things correctly (and I'm not sure I did), PCA/Wavelet is run on the whole dataset to obtain low dimensional representations of spikes. Then batches of N spikes are clustered. That sounds similar to what Kilosort does, except that batches are used for optimizing clusters rather than clustering them independently. What is the advantage of independent clustering? Mountainsort on the other hand follows a density based approach, which also seems to scale quite well with recording size. How does your method compare to a density based approach?

Author Response: The manuscript has been updated according to reviewers comment (Line 149 to 157).

11) Clustering accuracy: The measure you are using puts a higher weight on large clusters with a lot of spikes. In many datasets, these are multiunit clusters that are hard to separate. It would be nice to have some measure of temporal stationarity.

Author Response: The manuscript has been updated according to reviewers comment (Line 221 to 231).

Journal Requirements:

Author Response: The manuscript is according to the style requirements of PLOS One Journal.

2) Thank you for stating the following in the Acknowledgments Section of your manuscript:

[The research work is fully supported by Neural and Cognitive Systems Lab at Institute

for Intelligent Systems Research and Innovation, Deakin University.]

 [The author(s) received no specific funding for this work.]

Please include the updated Funding Statement in your cover letter. We will change the online submission form on your behalf.

Author Response: Funding related text is removed from the manuscript. We don’t require any updates in the funding statement.

---

## [Decision Letter · Decision Letter 1]

23 Dec 2019

PONE-D-19-23638R1

Efficient Neural Spike Sorting using Data Subdivision and Unification

PLOS ONE

Dear Dr. Bhatti,

Thank you for submitting your manuscript to PLOS ONE. After careful consideration, we feel that it has merit but does not fully meet PLOS ONE’s publication criteria as it currently stands. Therefore, we invite you to submit a revised version of the manuscript that addresses the points raised during the review process.

Please, seriously revise the manuscript to to clarify the concerns described  below and  fix  typos.

--Figure 7: Lines/symbols are overlapping to an extent that this figure becomes uninformative.

Maybe separate plots or cluster centroids for different segments? Please, provide a plot

showing the temporal stationarity of firing rates (for different segments).

--10) Temporal speedup: Please, clarify description of the algorithm concerning temporal speedup.

What is the advantage of independent clustering? How does your method compare to a

density based approach?

--11) Clustering accuracy: The measure you are using puts a higher weight on large clusters with a

lot of spikes. In many datasets, these are multiunit clusters that are hard to separate. It would

be nice to have some measure of temporal stationarity.

Other comments:

Please, clarify what readers are supposed to see in the example Figs 7+8 Both the conventional and the proposed method seem to produce identical results, but the sequence of plotting the different lines has changed. The bottom graphs are identical. Are these placeholder figures?

-'The surrounding region between −2SD to 2SD, containing about 95 percent of the

cluster data...' This statement is still wrong.

-What are 'Quirogo datasets'?

We would appreciate receiving your revised manuscript by Feb 06 2020 11:59PM. To enhance the reproducibility of your results, we recommend that if applicable you deposit your laboratory protocols in protocols.io, where a protocol can be assigned its own identifier (DOI) such that it can be cited independently in the future. For instructions see: http://journals.plos.org/plosone/s/submission-guidelines#loc-laboratory-protocols

We look forward to receiving your revised manuscript.

Kind regards,

Gennady Cymbalyuk, Ph.D.

Academic Editor

PLOS ONE

Reviewers' comments:

Reviewer's Responses to Questions

**Comments to the Author**

1. If the authors have adequately addressed your comments raised in a previous round of review and you feel that this manuscript is now acceptable for publication, you may indicate that here to bypass the “Comments to the Author” section, enter your conflict of interest statement in the “Confidential to Editor” section, and submit your "Accept" recommendation.

Reviewer #1: (No Response)

Reviewer #2: All comments have been addressed

2. Is the manuscript technically sound, and do the data support the conclusions?

Reviewer #1: No

Reviewer #2: Yes

3. Has the statistical analysis been performed appropriately and rigorously? 

Reviewer #1: I Don't Know

Reviewer #2: Yes

4. Have the authors made all data underlying the findings in their manuscript fully available?

Reviewer #1: Yes

Reviewer #2: Yes

5. Is the manuscript presented in an intelligible fashion and written in standard English?

Reviewer #1: No

Reviewer #2: Yes

6. Review Comments to the Author

Reviewer #1: I believe the authors still need more time to polish their manuscript. There are a lot of typos, and a few of my previous comments have not been addressed, specifically:

--Figure 7: Lines/symbols are overlapping to an extent that this figure becomes uninformative.

Maybe separate plots or cluster centroids for different segments? Please, provide a plot

showing the temporal stationarity of firing rates (for different segments).

--10) Temporal speedup: Please, clarify description of the algorithm concerning temporal speedup.

What is the advantage of independent clustering? How does your method compare to a

density based approach?

--11) Clustering accuracy: The measure you are using puts a higher weight on large clusters with a

lot of spikes. In many datasets, these are multiunit clusters that are hard to separate. It would

be nice to have some measure of temporal stationarity.

Other comments:

I'm still not sure what I'm supposed to see in the example Figs 7+8 Both the conventional and the proposed method seem to produce identical results, but the sequence of plotting the different lines has changed. The bottom graphs are identical. Are these placeholder figures?

-'The surrounding region between −2SD to 2SD, containing about 95 percent of the

cluster data...' This statement is still wrong.

-What are 'Quirogo datasets'?

Reviewer #2: The manuscript can be processed for publication as is. All comments have been addressed adequately

7. PLOS authors have the option to publish the peer review history of their article (what does this mean?). If published, this will include your full peer review and any attached files.

Reviewer #1: No

Reviewer #2: No

---

## [Author Response · Author response to Decision Letter 1]

3 Feb 2020

PONE-D-19-23638

Efficient Neural Spike Sorting using Data Subdivision and Unification

To the Editor,

Prof Gennady Cymbalyuk

We would like to acknowledge and appreciate the efforts and time of the editor and the reviewers for their invaluable comments and suggestions that has allowed us to enhance the quality of our manuscript. 

Below are the revisions according to valuable comments from the reviewers.

• Figure 7: Lines/symbols are overlapping to an extent that this figure becomes uninformative. Maybe separate plots or cluster centroids for different segments? Please, provide a plot showing the temporal stationarity of firing rates (for different segments).

Author Response: New figures (Fig 8 and Fig 09) are introduced clearly highlighting the performance difference between conventional and proposed algorithm in terms of computational efficiency and clustering accuracy. Separate cluster of different data segments are shown. Clustering outcome of intermediate steps is also shown to facilitate readers’ understanding of the proposed algorithm. 

• Temporal speedup: Please, clarify description of the algorithm concerning temporal speedup. What is the advantage of independent clustering? How does your method compare to a density based approach?

Author Response: A new figure (Figure 4) is added to address this comment, highlighting the differentiation between conventional spike sorting and proposed spike sorting mechanisms. Please refer to lines 132-140, and Table 1 for explanation on the effect of data size on the computational efficiency and temporal speedup of the clustering algorithms. In addition, please refer to lines 168-186 highlighting the comparison between density based approach and proposed method. 

• Clustering accuracy: The measure you are using puts a higher weight on large clusters with a lot of spikes. In many datasets, these are multiunit clusters that are hard to separate. It would be nice to have some measure of temporal stationarity.

Author Response: Please refer to accuracy index “A” highlighted in expression (4) and explanation in lines 290-307 where estimation of clustering accuracy is defined. Accuracy index is defined as the percentage of spikes accurately assigned to the relevant cluster, as per the ground truth, with respect to total number of spikes. This makes the accuracy index “A” independent of the number of spikes or the size of the cluster. As the contributing of the proposed mechanism is data processing rather than the clustering, which is adopted from conventional algorithms, therefore commenting on the temporal stationarity is out of the scope of this work. 

• 'The surrounding region between −2SD to 2SD, containing about 95 percent of the cluster data...' This statement is still wrong. 

What are 'Quirogo datasets'?

Author Response: The manuscript has been updated according to reviewers comment (Line 24 to 26). The reference of the adopted data set and brief explanation is also provide in lines 267-272. 

In addition the statement in reference to 2SD is updated, lines 246-247. 

Thanks 

Asim Bhatti

---

## [Decision Letter · Decision Letter 2]

7 May 2020

PONE-D-19-23638R2

Efficient Neural Spike Sorting using Data Subdivision and Unification

PLOS ONE

Dear Dr. Bhatti,

Thank you for submitting your manuscript to PLOS ONE. After careful consideration, we feel that it has merit but does not fully meet PLOS ONE’s publication criteria as it currently stands. Therefore, we invite you to submit a revised version of the manuscript that addresses the points raised during the review process.

We would appreciate receiving your revised manuscript by Jun 21 2020 11:59PM. To enhance the reproducibility of your results, we recommend that if applicable you deposit your laboratory protocols in protocols.io, where a protocol can be assigned its own identifier (DOI) such that it can be cited independently in the future. For instructions see: http://journals.plos.org/plosone/s/submission-guidelines#loc-laboratory-protocols

We look forward to receiving your revised manuscript.

Kind regards,

Alexandros Iosiﬁdis

Academic Editor

PLOS ONE

Additional Editor Comments (if provided):

Please address the comments of Reviewer 1.

Reviewers' comments:

Reviewer's Responses to Questions

**Comments to the Author**

1. If the authors have adequately addressed your comments raised in a previous round of review and you feel that this manuscript is now acceptable for publication, you may indicate that here to bypass the “Comments to the Author” section, enter your conflict of interest statement in the “Confidential to Editor” section, and submit your "Accept" recommendation.

Reviewer #1: (No Response)

Reviewer #2: All comments have been addressed

2. Is the manuscript technically sound, and do the data support the conclusions?

Reviewer #1: No

Reviewer #2: Yes

3. Has the statistical analysis been performed appropriately and rigorously? 

Reviewer #1: Yes

Reviewer #2: Yes

4. Have the authors made all data underlying the findings in their manuscript fully available?

Reviewer #1: Yes

Reviewer #2: Yes

5. Is the manuscript presented in an intelligible fashion and written in standard English?

Reviewer #1: No

Reviewer #2: Yes

6. Review Comments to the Author

Reviewer #1: Apart from a methological issue pointed out below (which needs to be discussed), a few missing details in the Methods and some awkward sentences and typos (I probably missed some and would encourage the authors to do another round of proofreading), this manuscript is now in a good shape.

Main points:

-please fix typos and grammatical errors (see below for a list of suggestions).

-I think I'm still missing some crucial information about the analysis. First I thought, that the performance improvement was somewhat related to nonstationarities in the data and you have shown (great, thanks) that this is clearly not the case. Another thing that I kept pointing out in my reviews and is still somewhat misleading in the presentation of the method is that in a high dimensional multivariate gaussian distribution, the probability for a datapoint to be within a 2 sigma radius from the center is not 95% but rather dependent on the number of dimensions, i.e. at most (95%)^d (for L1 norm), where d is the number of dimensions (or PCA components/ features). I haven't really found the number of dimensions you used in the paper (and you really do need to report it, it is a crucial number), but there is one figure suggesting the use of 10 features/dimensions. This seems high to me (and you may want to discuss such a parameter choice in the Discussion), what would have expected from other work would be 3-4 features. In any case, in the 10 feature case, your 2 sigma radius then accounts for at most 60% of the datapoints, so there are a lot of points outside your cluster boundaries. Does that explain why those widely used algorithms are working so poorly? If so, that's fine, but you want to discuss it in the Discussion section. It is also not clear to me how different dimensions are handled and you should elaborate a bit on that in the Methods. Is each dimension scaled such that variances match? If that is the case you're downweighting the first principal component and effectively explaining noisy, low variance features? Or am I missing something more subtle? You're reporting a performance improvement and I still don't see any reason why this should happen and especially why it would happen so consistently, given that all these algorithms have been used very successfully for years. I'm totally fine with the speed improvement and follow the argument that this should happen. But a general classification performance improvement is very hard to believe, so you need to at least report the specific circumstances under which it happens, i.e. the number of features/ dimensions and make clear that you're potentially inflating tiny differences in principal components with small variances (unless you corrected for that in some way, in that case it should be reported). Ideally, you should have some idea about a mechanism for the the performance improvement and discuss it in the Discussion (is it some kind of regularization effect that would be beneficial for noisy data?).

Specifically, do report the number of features/PCA components used. Do make clear whether the standard deviation was estimated for each component separately, thus enhancing the effect of small components, or whether (and how) you accounted for differences in the variances of features/PCA components. Ideally, specify a typical variation between variances of the features/PCA components (e.g. ratio between largest and smallest) and mention whether the results were sensitive to the number of PCA components.

A thorough analysis of the effect of dimensionality and scaling is certainly beyond the scope of this article, but I'm sure you made observations what happens if you change these parameters. You shall discuss them in the Discussion, and maybe even speculate about a mechanism or a scenario that tends to give performance improvements.

-Figure 7 has errorbars now, so please mention briefly how you obtained them/what they reflect. Further, numbers reported suggest a huge precision in comparison to these errorbars. Please round them, and wherever refered to in the text, add the uncertainty in brackets (e.g. 53+-6 %). You may leave the uncertainty in the table for clarity as it is already shown in Figure 7.

Other remarks

Figure 8+9: markers and labels don't match.

ln65: Brain consists

ln105: automatically estimate

ln119: presented data analysis issues due to progressive technological advancements of neural recordings

ln126: Although they have proposed an

efficient method for spike sorting, it still lacks the speed researchers require

ln130:The larger is the size the slower is the speed and large is the computational time required by spike sorting algorithms. -- rephrase or simply leave out (what, other than the obvious, are you trying to say?)

ln133: They reported, (?)

ln136: These second and third order operations prove the non-linear behaviour of spectral clustering.

ln138: To motivate our analysis,

ln141: The dependency of speed and computational time on data

size in spike-sorting has made it very difficult

ln142: identify the total number of

ln144: breast cancer cell data

ln149: Despite these challenges, ...

ln151: However, limited work has considered enhancing computational

ln153: The proposed algorithm pre-processes data to

ln154: time and to enhance speed and efficiency of a wide range

ln156: by parallel computing approaches to further

ln159: The novelty of the proposed mechanism

ln162:The first step involves subdivision of data into data-subsets of optimal length.

ln164:The second step involves clustering spikes in data-subsets

using conventional spike sorting algorithms.

ln165: The last step involves unification

ln166: clusters are then used to label

ln170:of conventional algorithms but rather performs additional data

ln171: the proposed mechanism very versatile and

ln175: uses a density based

The second step involves clustering

The last step involves unification

ln180: overall time of the spike sorting process.

ln193:The total number N of optimal subdivisions is estimated

ln195, 199: ,where L is the

ln223: of the algorithm depends on the length

ln227: (O L ) forms a direct

ln228: and an inverse relationship

the X-axis

the Y-axis

The computational time is the processing time after a movmean

filter (20 datapoints length) filtered the unwanted ripples in the plot and returned smooth curves. (representing computational time (why twice?))

The average value over ten repetitive analyses

robustness of the measure

ln241:'It is observed that, the

variations in data dimensionality does not have any effect on estimating the bounded

region. Whatever is the dimensionality of data, when the ED is calculated, the result is

always a single entry in one-dimensional space. For all EDs The standard deviation

(SD) is calculated using [66] and normal distribution curves are formed based on [67].' --Not even wrong. If you have a multivariate Gaussian distribution, the density distribution as a function of the radius is not Gaussian. The square of the radius (equals the sum of squares of Gaussian distributed random variables and) follows a Chi-squared distribution (check Wikipedia?) and you can imagine (take the cumulative distribution and rescale the x-axis) what follows for the distribution of the radius itself.

Figure 7: Continues positive trend is observed ???

Errorbars represent...

ln344: To cater for schocastic ??? variations of some of the algorithms

ln335: over 10 repetitons.

Reviewer #2: The latest revision and revised/new figures made the manuscript even more clear. The manuscript can be published as is.

7. PLOS authors have the option to publish the peer review history of their article (what does this mean?). If published, this will include your full peer review and any attached files.

Reviewer #1: No

Reviewer #2: No

---

## [Author Response · Author response to Decision Letter 2]

21 Jun 2020

PONE-D-19-23638-R3

Efficient Neural Spike Sorting using Data Subdivision and Unification

PLOS ONE

To the Editor,

Prof Alexandros Iosiﬁdis 

We would like to acknowledge and appreciate the efforts and time of the editor and the reviewers for their invaluable comments and suggestions that has allowed us to enhance the quality of our manuscript. 

Below are the suggested revisions according to valuable comments from the reviewers.

1) I think I'm still missing some crucial information about the analysis. First I thought, that the performance improvement was somewhat related to no stationarities in the data and you have shown (great, thanks) that this is clearly not the case. Another thing that I kept pointing out in my reviews and is still somewhat misleading in the presentation of the method is that in a high dimensional multivariate gaussian distribution, the probability for a datapoint to be within a 2 sigma radius from the center is not 95% but rather dependent on the number of dimensions, i.e. at most (95%)^d (for L1 norm), where d is the number of dimensions (or PCA components/ features).

Author Response: The manuscript is updated with additional information, mathematical expressions and references to address the points raised by the review as well as for the ease of the general readers (Line 237 to 252).

2) I haven't really found the number of dimensions you used in the paper (and you really do need to report it, it is a crucial number), but there is one figure suggesting the use of 10 features/dimensions. This seems high to me (and you may want to discuss such a parameter choice in the Discussion), what would have expected from other work would be 3-4 features.

In any case, in the 10 feature case, your 2 sigma radius then accounts for at most 60% of the datapoints, so there are a lot of points outside your cluster boundaries. Does that explain why those widely used algorithms are working so poorly? If so, that's fine, but you want to discuss it in the Discussion section. It is also not clear to me how different dimensions are handled and you should elaborate a bit on that in the Methods. Is each dimension scaled such that variances match? If that is the case you're down weighting the first principal component and effectively explaining noisy, low variance features? Or am I missing something more subtle? You're reporting a performance improvement and I still don't see any reason why this should happen and especially why it would happen so consistently, given that all these algorithms have been used very successfully for years. I'm totally fine with the speed improvement and follow the argument that this should happen. But a general classification performance improvement is very hard to believe, so you need to at least report the specific circumstances under which it happens, i.e. the number of features/ dimensions and make clear that you're potentially inflating tiny differences in principal components with small variances (unless you corrected for that in some way, in that case it should be reported). Ideally, you should have some idea about a mechanism for the performance improvement and discuss it in the Discussion (is it some kind of regularization effect that would be beneficial for noisy data?). Specifically, do report the number of features/PCA components used. 

Do make clear whether the standard deviation was estimated for each component separately, thus enhancing the effect of small components, or whether (and how) you accounted for differences in the variances of features/PCA components. Ideally, specify a typical variation between variances of the features/PCA components (e.g. ratio between largest and smallest) and mention whether the results were sensitive to the number of PCA components. A thorough analysis of the effect of dimensionality and scaling is certainly beyond the scope of this article, but I'm sure you made observations what happens if you change these parameters. You shall discuss them in the Discussion, and maybe even speculate about a mechanism or a scenario that tends to give performance improvements.

Author Response: Author Response: Further explanation is added, please refer to lines 282 to 298. 

3) Figure 7 has errorbars now, so please mention briefly how you obtained them/what they reflect. Further, numbers reported suggest a huge precision in comparison to these errorbars. Please round them, and wherever refered to in the text, add the uncertainty in brackets (e.g. 53+-6 %). You may leave the uncertainty in the table for clarity as it is already shown in Figure 7.

Author Response: Taking into account reviewer’s comment, Figure 7 is updated to provide simplified performance comparison. Performance outcomes, averaged over 10 repetitions, are presented for simplicity and ease of understanding. 

4) Other remarks

Figure 8+9: markers and labels don't match.

ln65: Brain consists

ln105: automatically estimate

ln119: presented data analysis issues due to progressive technological advancements of neural recordings

ln126: Although they have proposed an

efficient method for spike sorting, it still lacks the speed researchers require

ln130:The larger is the size the slower is the speed and large is the computational time required by spike sorting algorithms. -- rephrase or simply leave out (what, other than the obvious, are you trying to say?)

ln133: They reported, (?)

ln136: These second and third order operations prove the non-linear behaviour of spectral clustering.

ln138: To motivate our analysis,

ln141: The dependency of speed and computational time on data

size in spike-sorting has made it very difficult

ln142: identify the total number of

ln144: breast cancer cell data

ln149: Despite these challenges, ...

ln151: However, limited work has considered enhancing computational

ln153: The proposed algorithm pre-processes data to

ln154: time and to enhance speed and efficiency of a wide range

ln156: by parallel computing approaches to further

ln159: The novelty of the proposed mechanism

ln162:The first step involves subdivision of data into data-subsets of optimal length.

ln164:The second step involves clustering spikes in data-subsets

using conventional spike sorting algorithms.

ln165: The last step involves unification

ln166: clusters are then used to label

ln170:of conventional algorithms but rather performs additional data

ln171: the proposed mechanism very versatile and

ln175: uses a density based

The second step involves clustering

The last step involves unification

ln180: overall time of the spike sorting process.

ln193:The total number N of optimal subdivisions is estimated

ln195, 199: ,where L is the

ln223: of the algorithm depends on the length

ln227: (O L ) forms a direct

ln228: and an inverse relationship

the X-axis

the Y-axis

The computational time is the processing time after a movmean

filter (20 datapoints length) filtered the unwanted ripples in the plot and returned smooth curves. (representing computational time (why twice?))

The average value over ten repetitive analyses

robustness of the measure

ln241:'It is observed that, the

variations in data dimensionality does not have any effect on estimating the bounded

region. Whatever is the dimensionality of data, when the ED is calculated, the result is

always a single entry in one-dimensional space. For all EDs The standard deviation

(SD) is calculated using [66] and normal distribution curves are formed based on [67].' --Not even wrong. If you have a multivariate Gaussian distribution, the density distribution as a function of the radius is not Gaussian. The square of the radius (equals the sum of squares of Gaussian distributed random variables and) follows a Chi-squared distribution (check Wikipedia?) and you can imagine (take the cumulative distribution and rescale the x-axis) what follows for the distribution of the radius itself.

Figure 7: Continues positive trend is observed ???

Errorbars represent...

ln344: To cater for schocastic ??? variations of some of the algorithms

ln335: over 10 repetitons.

Author Response: The manuscript has been thoroughly revised taking into account all the comments by the reviewer. 

Thanks 

Asim Bhatti

---

## [Decision Letter · Decision Letter 3]

3 Aug 2020

PONE-D-19-23638R3

Efficient Neural Spike Sorting using Data Subdivision and Unification

PLOS ONE

Dear Dr. Bhatti,

Thank you for submitting your manuscript to PLOS ONE. After careful consideration, we feel that it has merit but does not fully meet PLOS ONE’s publication criteria as it currently stands. Therefore, we invite you to submit a revised version of the manuscript that addresses the points raised during the review process.

We look forward to receiving your revised manuscript.

Kind regards,

Alexandros Iosiﬁdis

Academic Editor

PLOS ONE

Journal Requirements:

Additional Editor Comments (if provided):

Please address the comments of Reviewer 1, by making sure that all definitions in the paper are precise.

Reviewers' comments:

Reviewer's Responses to Questions

**Comments to the Author**

1. If the authors have adequately addressed your comments raised in a previous round of review and you feel that this manuscript is now acceptable for publication, you may indicate that here to bypass the “Comments to the Author” section, enter your conflict of interest statement in the “Confidential to Editor” section, and submit your "Accept" recommendation.

Reviewer #1: (No Response)

2. Is the manuscript technically sound, and do the data support the conclusions?

Reviewer #1: Partly

3. Has the statistical analysis been performed appropriately and rigorously? 

Reviewer #1: Yes

4. Have the authors made all data underlying the findings in their manuscript fully available?

Reviewer #1: Yes

5. Is the manuscript presented in an intelligible fashion and written in standard English?

Reviewer #1: No

6. Review Comments to the Author

Reviewer #1: Lines 237-253: Please talk to a statistician (or someone who knows English and statistics) and reframe (and please put references from peer reviewed publications). This is about making your paper understandable to a reader, and I'm not asking for a layman explanation here (in fact, I feel that you're trying to explain a lot of things you don't need to explain, I'm asking for correctness. The standard deviation of a random variable is a well known and defined quantity and your equation does not reflect the standard deviation of euclidean distances.

The euclidean distances are strictly positive numbers, but in Figure 6, you're suggesting that they are Gaussian distributed, and therefore negative values are possible. So I really don't understand what you are doing here. And I would also be very interested whether you scale your principal components in some way, to match their variances, or whether the first principal components have larger weights.

The main issue that I raised in the last revision was that when you're working in a 10 dimensional space, things are a little more complicated. For example, if you have a standard normal distribution in 10 dimensions, then the euclidean distances (ED) follow a Chi-square distribution with 10 degrees of freedom (see https://en.wikipedia.org/wiki/Chi-square_distribution).

other comments:

ln. 288 kolmogorov-Smirnov (KS) test  Kolmogorov-Smirnov (KS) test

ln. 296 Please make clear what you mean by this sentence: 'It is

observed that 10 PCA features ensures the cumulative explained variance of over 85%

up to 95%, in case of the data sets employed in this study.' Please reframe this sentence.

The matlab function-- Please capitalize MATLAB.

7. PLOS authors have the option to publish the peer review history of their article (what does this mean?). If published, this will include your full peer review and any attached files.

Reviewer #1: No

---

## [Author Response · Author response to Decision Letter 3]

23 Sep 2020

To the Editor,

Gennady Cymbalyuk

We would like to acknowledge and appreciate the efforts and time of the editor and the reviewers for their invaluable comments and suggestions that has allowed us to enhance the quality of our manuscript. 

Below are the suggested revisions according to valuable comments from the reviewers.

1) Lines 237-253: Please talk to a statistician (or someone who knows English and statistics) and reframe (and please put references from peer reviewed publications). This is about making your paper understandable to a reader, and I'm not asking for a layman explanation here (in fact, I feel that you're trying to explain a lot of things you don't need to explain, I'm asking for correctness. The standard deviation of a random variable is a well-known and defined quantity and your equation does not reflect the standard deviation of Euclidean distances.

Author Response: The equation has been updated with a square term that was previously missing in the equation. (Please refer to Eq. 7.)

2) The Euclidean distances are strictly positive numbers, but in Figure 6, you're suggesting that they are Gaussian distributed, and therefore negative values are possible. So I really don't understand what you are doing here. 

Author Response: Detailed explanation of probability distribution of Euclidean distances is discussed at lines 273 to 318.

3) And I would also be very interested whether you scale your principal components in some way, to match their variances, or whether the first principal components have larger weights.

Author Response: The PCA components are not scaled to match their explained variances. The individual variances of PCA components are accumulated and the optimal number of PCA components that gives at least 85% of cumulative explained variance are chosen for the analysis. 10 PCA features are required to get at least 85% cumulative explained variance of the 64 dimensional spikes data. (Lines 353 to 358)

4) The main issue that I raised in the last revision was that when you're working in a 10 dimensional space, things are a little more complicated. For example, if you have a standard normal distribution in 10 dimensions, then the Euclidean distances (ED) follow a Chi-square distribution with 10 degrees of freedom (see https://en.wikipedia.org/wiki/Chi-square_distribution).

Author Response: The manuscript is updated with additional information, mathematical expressions, figures and references to address the points raised by the review as well as for the ease of the general readers. (Fig 6, Fig 7 and Lines 234 to 318).

5) Other comments:

i. ln. 288 kolmogorov-Smirnov (KS) test  Kolmogorov-Smirnov (KS) test

ii. ln. 296 Please make clear what you mean by this sentence: 'It is observed that 10 PCA features ensures the cumulative explained variance of over 85% up to 95%, in case of the data sets employed in this study.' Please reframe this sentence.

iii. The matlab function-- Please capitalize MATLAB.

Author Response: 

The manuscript has been updated according to reviewer comments.

Thanks 

Asim Bhatti

---

## [Decision Letter · Decision Letter 4]

10 Dec 2020

PONE-D-19-23638R4

Efficient Neural Spike Sorting using Data Subdivision and Unification

PLOS ONE

Dear Dr. Bhatti,

Thank you for submitting your manuscript to PLOS ONE. After careful consideration, we feel that it has merit but does not fully meet PLOS ONE’s publication criteria as it currently stands. Therefore, we invite you to submit a revised version of the manuscript that addresses the points raised during the review process.

We look forward to receiving your revised manuscript.

Kind regards,

Alexandros Iosiﬁdis

Academic Editor

PLOS ONE

Additional Editor Comments (if provided):

Please address the comments provided by the Reviewer on the current version of the paper, and submit a point-to-point response letter with the revised paper.

Reviewers' comments:

Reviewer's Responses to Questions

**Comments to the Author**

1. If the authors have adequately addressed your comments raised in a previous round of review and you feel that this manuscript is now acceptable for publication, you may indicate that here to bypass the “Comments to the Author” section, enter your conflict of interest statement in the “Confidential to Editor” section, and submit your "Accept" recommendation.

Reviewer #1: All comments have been addressed

2. Is the manuscript technically sound, and do the data support the conclusions?

Reviewer #1: Yes

3. Has the statistical analysis been performed appropriately and rigorously? 

Reviewer #1: Yes

4. Have the authors made all data underlying the findings in their manuscript fully available?

Reviewer #1: Yes

5. Is the manuscript presented in an intelligible fashion and written in standard English?

Reviewer #1: Yes

6. Review Comments to the Author

Reviewer #1: There are a few final edits I'd suggest in the modified part of the manuscript to improve readability, but I think the manuscript is in good shape now, and the methods are presented clearly.

lines 275-286 and Figure 7: I'd suggest to remove this paragraph and Figure 7 since

- I'm not sure how the explanation of a z-score helps in understanding the method.

- you are not plotting normal distributions in any of the Figures anymore, you defined the standard deviation in (7), so Equation (9) is not necessary (you're not using any other property of the normal distribution other than its standard deviation, and you cannot assume that the distribution of z scores is normal).

line 287: A short motivational sentence about what you're planning to do with the z score values would be great, e.g. 'We wanted to determine outliers for each spike cluster. To this aim, we considered two scenarios where the z scores distributions of a given cluster were either consistent with a normal distribution or skewed. There are numerous...'

if you rather want to keep these lines:

line 287: data distribution  z-score distribution

line 284: Euclidean

line 281, 283: 'used to plot', 'is plotted': I don't find these plots anywhere, so please reformulate.

7. PLOS authors have the option to publish the peer review history of their article (what does this mean?). If published, this will include your full peer review and any attached files.

Reviewer #1: No

---

## [Author Response · Author response to Decision Letter 4]

29 Dec 2020

1) lines 275-286 and Figure 7: I'd suggest to remove this paragraph and Figure 7 since

- I'm not sure how the explanation of a z-score helps in understanding the method.

- you are not plotting normal distributions in any of the Figures anymore, you defined the standard deviation in (7), so Equation (9) is not necessary (you're not using any other property of the normal distribution other than its standard deviation, and you cannot assume that the distribution of z scores is normal).

Author Response: As per reviewer’s suggestion paragraph (lines 275-286) and Figure 7 have been removed. 

2) line 287: A short motivational sentence about what you're planning to do with the z score values would be great, e.g. 'We wanted to determine outliers for each spike cluster. To this aim, we considered two scenarios where the z scores distributions of a given cluster were either consistent with a normal distribution or skewed. There are numerous...'

Author Response: The manuscript has been updated and a motivational sentence highlighting the use of Z scores is added at the start of the paragraph. (Please refer to lines 275-281).

3) if you rather want to keep these lines:

a. line 287: data distribution  z-score distribution. line 284: Euclidean

b. line 281, 283: 'used to plot', 'is plotted': I don't find these plots anywhere, so please reformulate. 

Author Response: We have removed the suggested text from the manuscript as per reviewer comments (1 and 2).

---

## [Editor Report · Decision Letter 5]

5 Jan 2021

Efficient Neural Spike Sorting using Data Subdivision and Unification

PONE-D-19-23638R5

Dear Dr. Bhatti,

We’re pleased to inform you that your manuscript has been judged scientifically suitable for publication and will be formally accepted for publication once it meets all outstanding technical requirements.

Kind regards,

Alexandros Iosiﬁdis

Academic Editor

PLOS ONE

Additional Editor Comments (optional):

Authors addressed all Reviewers' comments. Congratulations for the acceptance of your paper.
---

## [Editor Report · Acceptance letter]

21 Jan 2021

PONE-D-19-23638R5 

Efficient Neural Spike Sorting using Data Subdivision and Unification 

Dear Dr. Bhatti:

I'm pleased to inform you that your manuscript has been deemed suitable for publication in PLOS ONE. Congratulations! Your manuscript is now with our production department. 

Kind regards, 

on behalf of

Prof. Alexandros Iosiﬁdis 

Academic Editor

PLOS ONE